# Autophagy as a Potential Therapy for Malignant Glioma

**DOI:** 10.3390/ph13070156

**Published:** 2020-07-19

**Authors:** Angel Escamilla-Ramírez, Rosa A. Castillo-Rodríguez, Sergio Zavala-Vega, Dolores Jimenez-Farfan, Isabel Anaya-Rubio, Eduardo Briseño, Guadalupe Palencia, Patricia Guevara, Arturo Cruz-Salgado, Julio Sotelo, Cristina Trejo-Solís

**Affiliations:** 1Departamento de Neuroinmunología, Instituto Nacional de Neurología y Neurocirugía, Ciudad de México 14269, Mexico; eramed118@gmail.com (A.E.-R.); aaris16@gmail.com (I.A.-R.); phg6219@gmail.com (G.P.); patriciaguevara@yahoo.com (P.G.); tlacaelel333@gmail.com (A.C.-S.); jsotelo@unam.mx (J.S.); 2Laboratorio de Oncología Experimental, CONACYT-Instituto Nacional de Pediatría, Ciudad de México 04530, Mexico; racastilloro@conacyt.mx; 3Departamento de Patología, Instituto Nacional de Neurología y Neurocirugía, Ciudad de México 14269, Mexico; sergio.zavala.vega@gmail.com; 4Laboratorio de Inmunología, División de Estudios de Posgrado e Investigación, Facultad de Odontología, Universidad Nacional Autónoma de México, Ciudad de México 04510, Mexico; farfanmd@unam.mx; 5Clínica de Neurooncología, Instituto Nacional de Neurología y Neurocirugía, Ciudad de México 14269, Mexico; edubris@hotmail.com

**Keywords:** autophagy, glioma, chemotherapy

## Abstract

Glioma is the most frequent and aggressive type of brain neoplasm, being anaplastic astrocytoma (AA) and glioblastoma multiforme (GBM), its most malignant forms. The survival rate in patients with these neoplasms is 15 months after diagnosis, despite a diversity of treatments, including surgery, radiation, chemotherapy, and immunotherapy. The resistance of GBM to various therapies is due to a highly mutated genome; these genetic changes induce a de-regulation of several signaling pathways and result in higher cell proliferation rates, angiogenesis, invasion, and a marked resistance to apoptosis; this latter trait is a hallmark of highly invasive tumor cells, such as glioma cells. Due to a defective apoptosis in gliomas, induced autophagic death can be an alternative to remove tumor cells. Paradoxically, however, autophagy in cancer can promote either a cell death or survival. Modulating the autophagic pathway as a death mechanism for cancer cells has prompted the use of both inhibitors and autophagy inducers. The autophagic process, either as a cancer suppressing or inducing mechanism in high-grade gliomas is discussed in this review, along with therapeutic approaches to inhibit or induce autophagy in pre-clinical and clinical studies, aiming to increase the efficiency of conventional treatments to remove glioma neoplastic cells.

## 1. Introduction

About 2% of all malignant tumors are primary neoplasms of the central nervous system, with an incidence of 5–8/100,000 [1]. Gliomas, the most common brain tumors, can be classified as low-grade gliomas, anaplastic gliomas, or glioblastoma multiforme (GBM) [2]. GBM is the most aggressive and most vascularized type of solid tumor. It is characterized by a high mitotic activity, necrosis, inflammation, cellular proliferation, and thrombosis [3]. Glioblastomas may occur *de novo* as a grade-IV neoplasm (glioblastoma multiforme) or follow a malignant progression from low-grade (grade II) or anaplastic gliomas (anaplastic astrocytoma, grade III) to secondary gliomas [4]. Glioblastomas show an infiltrative growing pattern that makes them very resistant to surgery, radiotherapy, chemotherapy, or immunotherapy; in fact, patient survival time is as low as 12–15 months after diagnosis [5]. The resistance of GBM to a range of therapies is mainly due to a highly mutated genome and an overactivation of tyrosine kinase receptors, such as the epidermal growth factor receptor (EGFR), the platelet-derived growth factor receptor (PDGFR), and the vascular endothelial growth factor receptor (VEGFR), which have been found upregulated in GBM [5,6,7,8]. The stimulation of PDGFR, EGFR, and VEGFR by their ligands induces the activation of downstream signaling pathways, such as RAS-RAF-MAPK (including ERK, JNK, and p38) and PI3K-AKT-mTOR, which transduce signals to activate transcription factors, such as AP-1, NF-κB, Forkhead box class O (FOXO), HIF-1α, and β-catenin. These nuclear transcription factors regulate genes that are key for proliferation, cell cycle progression, apoptosis, autophagy, inflammation, angiogenesis, and invasion [9,10,11].

About 85% of GBM cases show an overregulation of the RAS/MAPK and PI3K/AKT pathways linked with the loss (37% of all GBM cases) or reduction (80% of all GBM cases) of the function of phosphatase and tensin homolog (PTEN). An increased expression of RAS and higher levels of RAS-GTP have been observed in several glioma cell lines and patient biopsies. In addition, the activation of RAS/RAF is due to the oncogenic mutations of *RAS* and *RAF* [9,10]. Genetic alterations of the malignant cells of GBM also involve the inactivation of tumor suppressor genes (*PTEN*, *P16*, *RB*, and *TP53*) [12,13,14], promoting cell proliferation due to a down-regulation of apoptosis by an increase in the levels of anti-apoptotic proteins (Bcl-2, Mcl-1, Bcl-xL, HIAP-1, HIAP-2, and XIAP) and a decrease in pro-apoptotic proteins (Bid, Bak, Bax, Bad, Bim, PUMA, NOXA, caspases-8, -10, -9, Apaf, DR4, Fas, and FADD) [15,16,17,18,19,20].

Due to a defective apoptosis in gliomas, promoting cell death by autophagy could be an alternative to remove tumor cells. Paradoxically, autophagy can promote either cell death or survival in neoplastic cells. The regulation of autophagy as a death mechanism in cancer cells has prompted the use of both autophagy inhibitors and inducers. It seems obvious that blocking autophagy can significantly enhance the sensibility of glioma cells to cytotoxic therapies and potentiate the effect of treatments in clinic trials [21,22]. On the other hand, a pharmacologically- or genetically-induced increase in autophagy has been linked to a significantly more efficient tumor removal in vitro and in vivo [23,24,25]. The role of autophagy as either a suppressor or an inducer carcinogenic mechanism in high-grade gliomas is discussed in this review, along with therapeutic strategies to inhibit or induce autophagy in pre-clinical and clinical studies, aiming to increase the therapeutic efficiency of conventional treatments to remove glioma neoplastic cells.

## 2. Pathways and Molecular Mechanisms of Autophagy

### 2.1. The Autophagic Process

Autophagy is a catabolic process that facilitates the recycling of cellular constituents in response to stressing conditions, such as nutrient deprivation or infection; thus, promoting the recovery of cellular balance. It has been classified into macroautophagy, microautophagy, and chaperone-mediated autophagy [26]. Macroautophagy involves training an autophagosome, a vesicle that will surround the proteins or organelles to be recycled and fuse with the lysosome to degrade the cargo; microautophagy involves invagination of the lysosome, with the target proteins or organelles inside it. In chaperone-mediated autophagy, the heat-shock cognate 70-kDa (HSC70) protein will recognize the KFERQ-motif in the target proteins and facilitate their translocation into the lysosome through the lysosomal-associated membrane protein 2A (LAMP2A) receptor [26]. This review will focus on macroautophagy, referred simply as autophagy from this point on.

Autophagy starts when proteins or organelles are encircled by the autophagosome, which then is fused with a lysosome. The fusion of an autophagosome and a lysosome (autolysosome) triggers the action of lysosomal enzymes that hydrolyze the content of autophagic vacuoles. Autophagy comprises the following stages: induction, nucleation, elongation, and completion; fusion to lysosome, demotion, and recycling [27]. The process involves a large complex of proteins that interact among them in response to inhibitory or activating conditions (Figure 1)

#### 2.1.1. Induction

In yeasts, autophagy induction requires the inhibition of the mammalian target of rapamycin complex 1 (mTORC1) and the activation of the canonical autophagy pathway, involving the *Atg* genes. In nutrient-rich media, mTOR activation leads to the hyperphosphorylation of Atg13 (mammalian homologue: ATG13), preventing thus its association to Atg1 (mammalian homologue: unc-51-like kinase 1 and 2 (ULK1 and ULK2)) and increasing its interaction with Atg11. During nutrient deprivation or treatment with rapamycin (mTORC1 inhibitor), Atg13 is hypophosphorylated, leading to the interaction between Atg1 and Atg13, triggering autophagy. Atg17 (mammalian homologue: FAK family kinase interacting protein, 200 kDa (FIP200)) is a protein that interacts with Atg13 and regulates the kinase activity of Atg1 [28]. It has been recently established that phosphorylated Atg17 is the basic protein required to form the phagophore assembly site (PAS), also known as omegasome in mammals. The formation of PAS is the point that actually marks the start of autophagy [29]. When Atg17 is located on the membrane, it acts as a recruiter protein to organize other Atg proteins, such as Atg11, Atg17, Atg20, Atg24, Atg29, and Atg31 [30,31,32] toward PAS [33]. Atg20 and Atg24 form a complex that interacts with Atg1, Atg18, Atg21, and Atg27 [34]. PKA inhibits autophagy by phosphorylating Atg1 and Atg13. PKA phosphorylates Atg1 in two different serine residues, and this step is required for Atg1 dissociation from PAS [35].

In mammals, autophagy is induced by the proteins ULK1/2; they are associated in a large complex with ATG13, FIP200, and ATG101, and are regulated by mTORC1. Under homeostatic conditions, mTORC1 phosphorylates and inhibits ULK1/2, but when nutrient deprivation occurs, mTORC1 is inhibited and dissociated from the ULK1/2 kinases, allowing ULK1/2 activation. The activated ULK1/2 kinases phosphorylate ATG13 and FIP200, causing the complex to relocate from the cytosol to the membrane of the endoplasmic reticle [36].

The process of relocation of ULK1 to the phagophore to initiate autophagy is not completely understood. Recently it was reported that the protein C9orf72, a guanine nucleotide exchange factor (GEF) [37], interacts with the Rab1/ULK1 complex, allowing its recruitment to the phagophore and mediating the initial step of autophagy. Low expression levels of C9orf72 are correlated with diseases such as amyotrophic lateral sclerosis and frontotemporal dementia, being an example of the importance of the regulation of the initial steps of autophagy [38].

#### 2.1.2. Nucleation

Several studies have suggested that nucleation takes place in the endoplasmic reticle in mammal cells. Autophagosome formation requires a vesicle to be formed through the Atg6 complex (mammalian homologue: Beclin-1), Vps34 (mammalian homologue: PI3K-III), Atg14, and Vps15 (mammalian homologue: p150) and the complex responsible for vesicle nucleation, which includes Atg1, Atg11, Atg13, and Atg17. The process of autophagosome formation is not well understood and new elements are integrated progressively. For example, the multispanning membrane protein Atg9 has been found to be essential for the autophagosome formation in yeast. Apparently, some Atg9 vesicles derived from the Golgi apparatus are able to assemble into PAS and become part of the autophagosomal outer membrane, with the possibility to be recycled later. In mammals, mAtg9 is highly conserved but, unlike yeast proteins, mAtg9 is only temporary part of autophagosomes and does not integrate to autophagosomal membranes [39].

Vesicle nucleation requires PI3K-III and Beclin-1 activation, along with additional factors recruiting proteins and lipids to form the autophagosome. The PI3K-III/Beclin-1 complex produces phosphatidylinositol 3-phosphate (PI3P), required to start the autophagosome formation [40]. PI3P binds proteins having PX or FYVE domains [41]. Atg20 and Atg24 (both of them important in forming pre-autophagic vacuoles) have PX domains to bind PI3P [42]. Recently, in a yeast model, the Atg2-Atg18 protein complex was also linked to the autophagosome formation. This complex is dependent on PI3P; in fact, Atg18 binds PI3P and PI3,5P_2_. Apparently, Atg2 tethers PAS and the ER to allow the formation and expansion of autophagosomal membrane, and its absence is linked to a minor rate of autophagy [43].

In addition, Beclin-1 has been reported to interact with the UV irradiation resistance-associated gene protein (UVRAG) [44], activating the autophagy/Beclin-1 regulator 1 (Ambra1) and the BAX-interacting factor-1 (Bif-1) [45]. UVRAG interacts with Beclin-1, and Bif-1 also interacts with Beclin-1 through UVRAG. These interactions induce the formation of the Beclin-1/PI3K-III complex, promoting autophagosome formation and maturation, since UVRAG stimulates the fusion of autophagosomes with lysosomes or late endosomes, prompting the degradation of their content in autophagic vesicles [44,45]. Several negative regulators of Beclin-1 have also been described. Such is the case of some members of the anti-apoptotic Bcl-2 protein family (Bcl-2 and Bcl-xL), which have four Bcl-2 homologue domains (BH) capable of inhibiting autophagy by interacting with Beclin-1. An example is NOXA, also a member of the Bcl-2 family, which constitutes a link between apoptosis and autophagy. NOXA binds MCL-1 and A1, allowing the release of BIM and activating apoptosis. However, when Beclin-1 displaces MCL-1, NOXA promotes autophagy. Reversely, autophagy promotes NOXA degradation through p62; and when autophagy is inhibited, NOXA accumulates and induces apoptosis [46].

The BH3 domain in Beclin-1 contributes to its association [47,48]. Bcl-2 is believed to block the interaction of Beclin-1 with PI3K-III, reduces the activity of PI3K-III, and decreases autophagy, either through the dissociation of Beclin-1/PI3K-III or by inhibiting their activity [47,49].

#### 2.1.3. Elongation and Completion

Autophagosome elongation and completion occurs through two different conjugation pathways: Atg8-Atg4 and Atg12-Atg5-Atg16. Both pathways regulate lipid (phosphatidylethanolamine, PE) conjugation to Atg8 (mammalian homologue: MAP-LC3). In the Atg12-Atg5-Atg16 pathway, Atg12 is activated by Atg7, which allows Atg10 to be transferred [50,51] and bind Atg5 [52]. The Atg12-Atg5 complex binds Atg16. This trimer oligomerizes and locates itself on the autophagosome outer surface. The Atg12/Atg5/Atg16 complex mediates the binding of LC3-PE to the autophagosome membrane [50,51]. For vesicle growth, the C-terminus end of MAP-LC3/Atg8 must be cleaved by the Atg4 cysteine-protease. This cleavage exposes a reactive glycine residue in the C-terminus end of MAP-LC3/Atg8, which is then activated by Atg3 and Atg7 and covalently bound to phosphatidylethanolamine (MAP-LC3-II-PE) on the autophagosome membrane. This action of Atg3 and Atg7 requires activating the conjugate Atg12-Atg5-Atg16. Recently, it was demonstrated that the expression of ATG7, ATG16L1, and LC-II is regulated by the RNA binding protein HuR after a hypoxic stimulus, proving the regulation of the ATG proteins in different stages and levels during autophagy, and the link between autophagy and hypoxia [53].

The recruitment of LC3-II to the autophagosome is the rate-limiting stage in the membrane expansion process. LC3-II mediates the closure of the membrane and its fusion with the lysosome [54]; the amount of LC3-II in PAS determines the autophagosome size [55]. Once the autophagosome is fully expanded, the LC3-II molecules anchored in the inner autophagosomal membrane are degradated [56], while LC3-II molecules on the autophagosome outer membrane are deconjugated from PE by Atg4 and released back into the cytosol [57]. Similarly, the Atg12-Atg5-Atg16 complex is dissociated from the autophagosome membrane during maturation.

With respect to the closure mechanism of the autophagosome, CHMP2A, a component of the endosomal sorting complexes required for transport (ESCRT)-III was recently identified as a regulator of the phagophore closure. These complexes, described as ESCRT-0, -I, -II, -III, and Vps4, are involved in the sorting of ubiquitinated cargo into multivesicular bodies, in cytokinesis and in the budding of enveloped viruses, such as HIV [58]. CHMP2A, one of the four core subunits of ESCRT-III, plays a key role in the separation of the inner and outer autophagosomal membranes, and therefore it is required for a functional autophagosome [56]. Similarly, the depletion of CHMP2A complexes results in the accumulation of unclosed autophagosomes, indicating the importance of these proteins in autophagy [59].

#### 2.1.4. Fusion with Lysosomes and Degradation

After maturation, autophagosomes are fused to lysosomes to form the autophagolysosome, which requires the action of several proteins, such as Rab7 and Vti1p in mammalian cells [60,61].

In mammals, the fusion of autophagosomes to lysosomes depends on the soluble NSF attachment protein (SNARE) receptor, being syntaxin 17 (Stx17) a key protein in this step. After its recruitment to autophagosomes, Stx17 forms a trans-SNARE complex to allow membrane fusion. The recruitment of Stx17 to the autophagosomes is crucial. Additionally, a direct interaction of Stx17 with the small guanosine-triphosphatase named as immunity-related GTPase M (IRGM) and of both with Atg8 proteins seems to be required for autophagosome assembly; this complex, named as autophagosome recognition particle (ARP), is critical in this early step of autophagolysosome formation [62].

UVRAG plays another role in later autophagy stages: it regulates autophagosome maturation independently of Beclin-1. UVRAG facilitates the recruitment of the C vacuolar protein (C-VPS) to the autophagosome. The ensuing autophagosome-lysosome fusion leads to a rupture of the inner membrane and a degradation of the cytosolic content by lysosomal hydrolases [50,51]. The interaction of UVRAG with the C-VPS complex promotes the activity of a Rab7-GTPase; in conjunction with proteins, such as LAMP-1 and LAMP-2, this results in the fusion of the autophagosome and lysosome [63]. The material to be engulfed by autophagosomes is selected by protein interactions with specific autophagy receptors. These proteins have LC3-II-interaction domains named as LC3-interacting regions (LIR); they also feature an ubiquitin-recognition domain at the C-terminus end named as ubiquitin-associated (UBA) domain, which is able to bind mono- or poly-ubiquitinated regions, and a PB1 oligomerization domain at the N-terminus end. The best example of these proteins is p62, or sequestosome-1 (p62/SQSTM1). This protein is able to recognize ubiquitinated proteins through the UBA domain, as well as misfolded proteins and varying-size protein aggregates through the oligomerization domain. Once these aggregates are formed, they are recognized by LC3-II on the autophagosome inner face and recruited inside [64]. Another protein recently reported as driving degradation of protein aggregates is TRIM16, which acts by interacting with p62, the Kelch-like ECH-associated protein 1 (KEAP1) and the nuclear factor erythroid 2-related factor 2 (NRF2) [65]. Similarly, dysfunctional organelles such as depolarized mitochondria can be recognized, in a process in which only participate BH3-family proteins, such as BNIP3 and BNIP3L [64].

### 2.2. AMPK and mTOR in Autophagy Regulation

The main autophagy regulators are the 5′-AMP-activated protein kinase (AMPK) and the serine/threonine protein-kinase (mTOR) (Figure 1). Both kinases respond to significant nutritional alterations. mTOR acts as an intracellular sensor of the cell energetic status, regulating cell processes, such as proliferation, survival, and energetic metabolism [36]. In the presence of amino acids and growth factors, mTOR regulates protein synthesis through the phosphorylation and subsequent activation of the ribosomal subunit S6 kinase (p70^S6K^), which induces protein synthesis by translating transcripts coding for elongation factors and ribosomal proteins. On the other hand, mTOR induces phosphorylation and inactivation of the elF4 factor inhibitor (4E-BP1), allowing its activation and starting translation [36]. mTOR has been reported to be inactivated by a lack of nutrients, growth factor deprivation, rapamycin administration, or stress. These factors inhibit protein synthesis and activate autophagy to obtain energy [66].

In mammals, mTOR is composed by two multi-protein complexes, TORC1 and TORC2. The TORC1 complex primarily regulates cell growth, energetic metabolism, and autophagy; it is sensitive to rapamycin and it is constituted by mTOR, mLST8, and Raptor [67,68]. TORC2 is formed by mTOR, mLST8, rictor, and protor [68,69]. It is not sensitive to rapamycin, and it regulates cell proliferation and cytoskeleton reorganization [68]. The TORC1 complex is mainly regulated by AMPK (inhibitor), ERK, and AKT (activators) (Figure 1). The presence of growth factors (EGF and PDGF) leads to the formation of PIP3 via PI3K I, with the ensuing activation of AKT. Growth factors also lead to the activation of ERK via RAS/MEK/ERK. ERK and AKT induce phosphorylation of TSC2, preventing the TSC2/TSC1 complex to be formed and the activation of TORC1. TORC1 blocks the initiation of autophagy through the phosphorylation (inactivation) of Atg13, ULK, AMBRA, and Atg-14L, inhibiting the complex Atg13-ULK1 and the activity of ULK1 and vps34. mTOR also phosphorylates WIP12, with its subsequent ubiquitination and proteosomal degradation via the E3 ubiquitin ligase HUWE1 [70]. Besides, mTOR inactivates the autophagic process through the phosphorylation (Ser-142 and Ser-211) and cytosolic retention of the transcriptional factor TFEB, which regulates de transcription of genes related to lysosome formation and autophagy such as *UVRAG*, *LC3-II, VPS11, P62/SQSM1*, and *WIP*. TFEB phosphorylated in Ser-211 is recognized and sequestered into the cytoplasm by 14-3-3 phospho-proteins. TSC1/TSC2 acts as a GTPase of Rheb, a GTP-binding protein that activates TORC1 [71,72] (Figure 1). In yeasts, TORC1 inhibits autophagy by hyperphosphorylating atg13, thus inhibiting its interaction with atg1 [30,32]. Apparently, alterations in nutrient levels induce changes through mediators, such as microRNAs. In glioblastoma, miR-451 is expressed in the presence of high levels of glucose and upregulates the LKB1/AMPK/mTOR pathway, which finally leads to proliferation [73,74].

AMPK also acts as a bioenergetic sensor. During nutrient or energy deprivation, AMPK is activated by a decrease in the ATP/AMP ratio via LKB1. LKB1 is able to phosphorylate threonine-172 in AMPK, activating it. On the other hand, when intracellular levels of calcium and some cytokines (TRAIL, TNF or IL-1) increase, the Ca^2+^/calmodulin-dependent kinase kinase β (CaMKKβ) and the transforming growth factor-β-activating kinase 1 (TAK1) are activated; both CaMKKβ and TAK1 are able to activate AMPK. AMPK induces autophagy by phosphorylating TSC1/TSC2 and inactivating mTOR [75,76]. Interestingly, AMPK can also regulate autophagy through upstream mediators, such as the death-associated protein kinase 2 (DAPK2). AMPK can phosphorylate DAPK2, inducing a conformational change that reduces its homodimerization and also enhances the phosphorylation of Beclin-1 by DAPK2; thus, the decrease of the DAPK2 function caused by AMPK activation results in a reduced rate of autophagy [77].

The interaction between mTOR and AKT in autophagy could be exploited as a therapeutic target. For example, Temozolomide (TMZ), a novel drug used in glioblastoma, activates AMPK and AKT, but only AMPK leads to apoptosis, while AKT activation could be linked with a resistance to the treatment. Cordycepin, another new drug, has a synergistic effect with TMZ. Cordycepin seems to activate AMPK as well, while suppressing AKT [78]. In a similar strategy, coinfection with the LaSota strain of the Newcastle disease virus (NDV-LaSota) enhances the effectiveness of TMZ, since NDV inhibits AKT but activates AMPK [79].

Additionally, metformin has been reported to increase the expression of AMPK while inhibiting the expression of mTOR, leading to cell death in glioma models [80]. In a different approach, it has been reported that specific antigens, such as the melanoma antigen-6 (MAGE) are related with the degradation of the tumor-suppressing protein AMPK; when *MAGE* is knocked-out, AMPK activity is restored, and mTOR is inactivated, leading to cell death by autophagy [81].

In glioblastoma, autophagy induction could be used as a complementary therapy. In this context, omega-3-polyunsaturated fatty acids (ω3-PUFAs), such as docosahexaenoic acid (DHA), have been reported to increase autophagy by activating AMPK and dephosphorylating AKT and mTOR [82].

### 2.3. Regulation of Autophagy at a Nuclear Level by Transcription Factors

Autophagy can also be regulated by transcription factors, such as p53, FOXO, and TEFB, which can either increase or suppress the expression of proteins involved in this process.

#### 2.3.1. P53

The tumor-suppressing gene *TP53* is activated by conditions, such as DNA damage or oxidative stress. In the nucleus, p53 regulates the expression of proteins that define the fate of the cell: cell-cycle arrest or apoptosis, but also autophagy [83]. P53 seems to promote the autophagic processes by enhancing the expression of genes involved in autophagy induction (*LKB1*, *ULK1/2*) and autophagosome maturation (*ATG4*, *ATG7*, and *ATG10*). It also activates the expression of Sestrin, a protein that activates AMPK; this kinase phosphorylates the complex TSC1,2 and inhibits mTOR, thus activating autophagy [84]. p53 also increases the expression of the damage-regulated autophagy modulator (DRAM), which activates autophagy and apoptosis in a concerted manner [85].

In cancer, a mutated p*53* decreases autophagy and favors the proliferation of tumoral cells, and thus it could be a therapeutic target. In autophagy, anti-apoptotic proteins, such as Bcl-xL and XIAP are also degraded, thus promoting the removal of tumor cells by cytotoxic lymphocytes and NK cells through the Granzyme pathway [86]. In fact, p53 activates autophagy; conversely, autophagy suppresses the expression of p53, thus promoting cancer proliferation [87].

#### 2.3.2. TFEB

The transcription factor EB (TFEB) is linked primarily to lysosome biogenesis, but it also modulates the activation of autophagy in starvation [88]. mTOR, ERK2, and GSK3B are the main kinases that phosphorylate and sequester TFEB in the cytoplasm [89,90]; but when TFEB is dephosphorylated, it is translocated to the nucleus and controls processes, such as autophagosome formation and autophagosome-lysosome fusion. In fact, when ERK2 and GSK3B phosphorylate TFEB, they inhibit the nuclear translocation of and thus decrease autophagy [90]. The genes *UVRAG, WIPI, MAPLC3B, SQSTM1, VPS11, VPS18*, and *ATG9B* have a TFEB target site in their promoters; thus, their expression increases in correlation with TFEB [90]. It has been reported that TFEB is overexpressed in glioblastoma, along with other genes that codify for autophagic proteins, such as LC3A, LC3B, Beclin 1, Ulk 1, Ulk 2, p62 [91], and SQSTM1 [92]. Interestingly, the expression of SQSTM1 in glioblastoma cells is increased with the inhibition of the WNT pathway, sensitizing tumor cells to the effects of autophagy inhibitors and thus could offer an alternative therapy for this cancer type [92].

#### 2.3.3. FoxO

The Forkhead box class O (FoxO) family of transcription factors is another autophagy regulator. Four family members have been described in mammals: FoxO1, FoxO3, FoxO4, and FoxO6 [93]. FoxO proteins are usually located in the cytoplasm and translocate to the nucleus to induce the expression of several genes, including those that regulate autophagy induction (*ULK1*, *ULK2*, *SESN3*), nucleation (*BECN1*, *ATG14*, *PI3KIII*), elongation (*MAP1LC3B*, *ATG4*, *ATG5*, *ATG12*, *GABARALI*), and autophagosome-lysosome fusion (Rab7 and TFEB, another transcription factor) [94]. FoxO can also modulate autophagy at different levels by interacting with autophagy regulators, such as Atg7 [95], or through epigenetic mechanisms, such as histone modification [96] and microRNAs. Conversely, autophagy can regulate FoxO expression.

#### 2.3.4. HIF

Autophagy can also be induced by hypoxia, and HIF-1α is key in this process. Under normoxic conditions, HIF-1α is hydroxylated by the prolyl hydroxylase domain protein 2 (PHD2), and then it interacts with the von Hippel Lindau protein (VHL). Later, the E3 ubiquitin ligase is recruited to the complex, and HIF-1α undergoes further degradation in the proteasome. Conversely, under hypoxic conditions, HIF-1α is stabilized and accumulated in cytoplasm, and it is translocated to the nucleus for the transcription of several genes involved in metabolism, proliferation, and angiogenesis [97]. HIF-1α induces mitochondrial autophagy (mitophagy) linked with the expression of BINP3, BNIP3L, Beclin-1, and Atg5 [98]. BINP3 also competes with Beclin-1, impairing the interaction of Beclin-1 with Bcl-2 [99] and releasing Beclin1 to activate autophagy [100]. This mechanism helps the cell to survive under hypoxic conditions. In cancer, autophagy has a dual role: in a first stage, it helps tumor cells to adapt to an hypoxic microenvironment; however, autophagy induction could favor the removal of tumor cells. Interestingly, Beclin1 is considered as a tumor-suppressing gene and as an autophagy initiator; its deficiency, linked with higher angiogenesis and cell proliferation in cancer, is more pronounced under hypoxic conditions, with a consistent increase in the levels of HIF-2α, but not HIF-1α [101]. In contrast, it has been reported that Beclin1 is linked to cell survival through hypoxia-induced autophagy in cancer, and a downregulation of Beclin1 under hypoxic conditions leads to cell death [98].

However, HIF-1α it is not the only regulator of autophagy under hypoxic conditions; AMPK has been also proposed to trigger autophagy in hypoxia [102]. Another example are the mutations inactivating the tumor-suppressing gene *VHL*, that correlate with a higher rate of angiogenesis and tumor cell survival in renal cancer. When the function of *VHL* is restored with synthetic molecules, an increase of autophagy is observed independently of HIF-1α [103].

Autophagy counteracts the effects of hypoxia, regulating the levels of HIF-1α. In this context, HIF-1α interacts with HSC70 and LAMP2A, both components of the chaperone-mediated autophagy (CMA) complex, and then it is degraded in autophagic lysosomes [104].

#### 2.3.5. PTEN

Another transcription factor that activates autophagy is the phosphatase and tensin homolog (PTEN). Chemotherapeutic agents, such as cisplatin and topoisomerase I inhibitors cause DNA damage and as a result activate ATM by phosphorylation. Then, ATM phosphorylates PTEN, allowing its nuclear translocation to induce autophagy by inhibiting the AMPK pathway [105]. Thus, autophagy has been proposed to favor the survival of tumor cells.

#### 2.3.6. E2F1 and NF-κB

NF-κB downregulates the expression of BNIP3 in normoxia; however, under hypoxic conditions, NF-κB is released from the BNIP3 promoter and allows the action of E2F1 to induce autophagy, which also promotes the expression of the autophagic genes *ULK1*, *LC3*, and *ATG5* [106].

## 3. Autophagy in Glioma

The effect of autophagy in glioblastoma during tumor initiation, promotion, and progression is still controversial, given its regulatory role in both cell death and survival. Autophagy has been proposed as an inhibitor of tumor initiation by removing damaged proteins and organelles, protecting cells from reactive oxygen species (ROS), necrosis, inflammation, genomic instability, and metabolic alterations [107]. However, it has been proposed that glioma cells promote autophagy under adverse circumstances, such as nutrient deprivation, acidosis, oxidative, or hypoxic stress, to sustain their survival, evading the physiological response to cancer and therapy [108,109] (Figure 2). Autophagy could impact on the prognosis of glioma, either positively or negatively.

### 3.1. Autophagy as a Tumor Suppressor in Glioma

It has been demonstrated that autophagy inhibits the tumor initiation stage, removing cancer cells during tumor progression. Autophagy decreases the proliferation of MCF-7 breast tumor cells overexpressing Beclin-1, as well as the incidence of spontaneous tumors in Beclin-1-haplodeficient mice [110]. Astrocytoma and glioblastoma cells show lower levels of autophagy-related proteins with respect to low-grade astrocytic tumors [111,112,113]; on the other hand, it has been demonstrated that the progression of astrocytic tumors is associated with a decrease in autophagic capacity [114]. Shukla et al. reported that ULK1 and ULK2 mRNA and protein levels are significantly decreased in glioblastoma patients with respect to normal brain samples, promoting astrocytic transformation by impairing autophagy [115]. Additionally, a lower expression or deletion of important genes for autophagosome initiation and elongation, such as *FIP200*, *Beclin-1*, *UVRAG*, *Bif1 Atg4c*, and *Atg5* has been reported in GBM [115]. Lower levels of Beclin-1 transcript and protein were reported in glioblastoma [116]. It is noteworthy that the increased levels of the proteins LC3 and Beclin-1 were associated with an improved survival in GBM patients with poor performance scores [117,118]. A more intense hyperphosphorylation (activation) of AKT and mTOR has been reported in grade-III and -IV gliomas than in low-grade gliomas [119] as related with a poorer prognosis in glioblastoma patients [120].

In addition, the activation of the mTOR signaling pathway correlates with an inhibition of the autophagic process, favoring the proliferation and pluripotency of glioma stem cells [121]. Similarly, glioma stem cells promoted therapeutic resistance, tumor infiltration, treatment, failure, and relapse [122]. It has been proposed that a drop in the expression of Beclin-1 in GBM tissue enhances EGFR overexpression; therefore, higher EGFR levels and decreased levels of the protein Beclin-1 have been associated with tumor progression and a poorer prognosis [123]. Wang et al. reported that a high miR-33a expression correlates with a poor prognosis in glioblastoma patients by blocking the tumor suppressor protein UVRAG [124]. miR-224-3p is downregulated in glioblastoma cells under hypoxic conditions; its expression in glioblastoma tumors is low, as expected in very hypoxic tumors. miR-224-3p inhibited autophagy by suppressing ATG5 and FIP200, and its overexpression inhibited tumorigenesis in glioblastoma cells [125]. BNIP3, a pro-cell death Bcl-2 family member, is upregulated in hypoxia, inducing autophagy in glioma cell lines [126]. Correspondingly, higher BNIP3 levels were found in patients who failed to respond to a VEGF-neutralizing antibody [127].

Autophagy may exert its tumor-suppressing function in various ways (Figure 2). Autophagy may protect genomic integrity by removing damaged organelles, which could produce reactive oxygen species (ROS) and cause genomic instability, altering genic expression and increasing the rate of chromosome gain and loss, thus inducing tumor progression [128]. Furthermore, autophagy may suppress tumorigenesis by removing p62-tagged aggregates. p62 accumulation causes DNA, protein, and mitochondrial damage and the generation of ROS, that promote instability genomic and tumor progression [129]. An overexpression of p62 has been reported in GBM patients, correlated with a poorer prognosis [118]. p62 inhibited autophagosome and protein degradation of Twist1, favoring the epithelial-mesenchymal transition (EMT) [130]. Furthermore, p62 enhances RAS-activated cellular transformation by activating NF-κB [131]. It has been proposed that p62, via PKC and/or the E3 ubiquitin ligase TRAF6, induces the activation of NF-κB by phosphorylating IκBα, leading to its ubiquitination and proteasomal degradation, to finally induce the translocation of NF-κB from cytosol to the nucleus [132,133]. Jing et al. demonstrated that sulfasalazine inhibited the NF-κB pathway through the autophagic degradation of p62, and thus inhibited the proliferation of U251 glioma cells [134]. In addition, it has been reported that a decreased expression of p62/SQSM1 significantly decreases ERK phosphorylation, attenuating the proliferation and invasion of glioma cells induce by Guanylate binding proteins-3 (GBP) in vitro [135]. Additionally, p62 has been reported to inhibit apoptosis by blocking the activation of caspase-7, which participates in the execution steps of apoptosis [136]. Another mechanism by which autophagy can prevent the carcinogenic process involves inhibiting or reverting EMT. Autophagy activation is associated with the degradation of Snail, a transcription factor that regulates EMT [137]. EMT and its reverse process (MET) are both essential for cancer migration, invasion, chemo- and radioresistance [138]. EMT allows tumoral cells to be released from the primary tumor and invade the brain parenchyma and blood vessels [137]. Jiang et al. demonstrated that Sinomenine hydrochloride (SE) induces cell death by autophagy in glioma cells through the generation of ROS, with the ensuing inhibition of the AKT/mTOR pathway and activation of JNK [139]. Furthermore, it has been reported treating U87 and SF767 glioma cells with SE induced cell cycle arrest in the phase G_0_/G_1_ by decreasing the expression of the cyclins D1, D3, and E, and by increasing p21 and p27 (CDK inhibitors). SE also inhibits migration and invasion by repressing NF-κB, and in consequence decreasing the expression of MMP2/9, as well as reversing EMT by inducing autophagy caused by endoplasmic reticulum (ER) stress [139]. In addition, those authors confirmed the inhibition of EMT by a downregulation of mesenchymal markers, such as vimentin (intermediate filament protein), SNAIL, and Slug, as well as decreased invasion rates of glioma cells to the inflammatory microenvironment [139]. Autophagy induced by amino acids and rapamycin in GL15 and U87 glioma cells was reported to revert EMT, reducing tumor cell migration and invasion [140], by inhibiting the synthesis of SNAIL and increasing the expression of N-cadherin and R-cadherin. Those authors suggested that autophagy modulation triggers a molecular switch from a mesenchymal phenotype to an epithelial-like one in cellular GBM. On the other hand, SNAIL and TWIST are degradated by the autophagy-lysosome degradation machinery in cancer cells upon PI3K or Beclin-1 overexpression, and vice versa [137].

Autophagy also inhibits malignant transformation by mediating senescence [141]. Cellular senescence is a steady status of proliferative detention that limits malignant transformation. It has been reported that TMZ induced autophagy followed by senescence in glioma cells [142]. Filippi-Chiela et al. reported that resveratrol enhances the antineoplastic effect of TMZ, resulting in a mitotic arrest followed of senescence in glioma cells [143]. Yuan et al. also proved that resveratrol enhanced the toxicity of TMZ by increasing ROS production, which induce the activation of AMPK, followed by the inhibition of the mTOR pathway and a decrease in Bcl-2 [144]. Flovokawain (chalcone) arrested cellular proliferation in the U251, T98, and U87 GBM lines by autophagy activation followed by senescence mediated by ER stress; it also inactivated the AKT/mTOR pathway [145]. Matrine (an alkaloid extract from *Sophora flavescens*) potently inhibits growth and invasion in GBM lines, inducing senescence by inactivating the IGF1/PI3K/AKT/p27 pathway [146].

Another mechanism by which autophagy may block tumor formation and progression is the induction of apoptosis via the protein ATG [147,148]. It has been suggested that Beclin-1 could exert its pro-apoptotic effect by preventing the anti-apoptotic function of Bcl-xL and Bcl-2. Huang et al. showed that Beclin-1 elicits apoptosis by binding Bcl-2 and Bcl-xL, thus releasing Bax and Bak, which activate caspases-3/-9 in glioma cells [149].

It has been proposed that atg5 is a molecular mediator between autophagy and apoptosis. An N-terminal fragment is cleaved from atg5 by calpain 1 and 2; the 24-kD truncated atg5 is translocated to the mitochondria, where it inhibits Bcl-xL, enhancing the activation of Bax, with the ensuing activation of caspase-9 [150]. Radiation (10 Gy) on glioma cell lines on days 3 and 5 induced autophagy prior to apoptosis, while *atg5*-knockdown in U373 and LN229 glioma cells after radiation significantly decreased both autophagy and apoptosis, independently of caspase activation, suggesting that atg5 is required for apoptosis induction [151].

Furthermore, it has been demonstrated that p62 promotes apoptosis by activating caspase-8 [152]. Zhang et al. reported that HAMLET (a decalcified a-lactalbumin and oleic acid complex) induces the activation of p62, leading to cell death by apoptosis in the U87 glioma line by activation of caspase-8 [153]; those authors suggested that ubiquitinated caspase-8 could activate the autophagosome by binding p62 [153,154]. The atg12-atg3 complex induces apoptosis through the mitochondrial pathway [155]. atg12 binds and inactivates the anti-apoptotic proteins Bcl-2 and Mcl-1, leading to BAX activation, the permeabilization of the outer mitochondrial membrane, cytosolic release of cyt c, and activation of apoptosis-executing caspases [156]. Another autophagic protein that regulates cell death by apoptosis is the cysteine-protease atg4D, which is hydrolyzed by caspase 3. A truncated atg4D is produced as a result, which is translocated to mitochondria with a “BH3”-like domain exposed in the C-terminus end that allows it to bind Bcl-2 family members and induce apoptosis [157].

### 3.2. Autophagy as a Tumoral Promotor in Glioma

It has been demonstrated that the induction of autophagy by stress in tumor cells can result in resistance to the treatment, with the consequent tumor recurrence and progression [158]. Noor et al. reported that inhibiting autophagy by genetically suppressing Atg7, Atg13, or ULK1 significantly decreased glioma growth and oncogenic progression in a KRAS-driven GBM mouse model, suggesting that autophagy is essential for the initiation and growth of glioma [159].

Autophagy can be associated with the progression of glioma, especially in high-grade glioma. In fact, the levels of LC3 and p62 significantly correlated with a poorer prognosis, suggesting that LC3 and p62 could be considered as useful prognostic factors of glioma [160]. Additionally, an overexpression of autophagic-related proteins has been observed in a high proportion of glioblastoma patients, with a significant increase of ULK1/ULK2 and TFEB [91]. A strong upregulation of p62 and LAMP2 was detected in glioblastoma peri-necrotic areas, suggesting that microenvironmental changes act as a driver of autophagy induction in gliomas [161]. A higher DRAM and p62 expression has been observed in glioblastoma from adult patients, and both proteins were highly correlated with a poorer prognosis [118]. DRAM and p62 induce migration and invasion in glioblastoma stem cells through a metabolic de-regulation via RAS/MAPK [118]. Tamrakar et al. demonstrated that a marked increase in the expression of p62, LC3, and Beclin-1 was related with radiation therapy in glioblastoma biopsy samples, whereas LC3 and p62 expression was associated with a poorer overall survival, and LC3 was associated with the methylation in the promoter of O^6^-methylyguanine-DNA methyltransferase (MGMT) and telomerase reverse transcriptase (TERT). p62 showed a strong association with 1p/19q co-deletion, IDH1 mutation, and MGMT promoter methylation, while Beclin-1 showed a strong association with isocitrate dehydrogenase 1 (IDH1) mutation and 1p/19q co-deletion [160].

On the other hand, a higher expression of LC3/Beclin-1 correlated with a shorter progression-free survival in low- and high-grade glioma patients [162]. Wen et al. found that a higher ATG4C transcript expression in patients with high-grade glioma correlated with a reduced overall survival (OS). ATG4C knockdown in T98G glioma cells inhibited autophagy, enhancing the cell cycle arrest and promoting apoptosis through the production of ROS, the expression of p21, p53, and Bax, and decreased levels of Bcl-2 [163]. A drop in the expression of ATG4C improved the sensitivity of U87-MG and T98G glioma cells to TMZ by inhibiting autophagy. Furthermore, ATG4C KO significantly reduced glioma growth rates in nude mice [163]. By quantifying autophagosomal molecules, such as LC3B, p62, BAG3, and Beclin-1, it was proved that oxygen or nutrient deprivation increase autophagy in astrocytoma with respect to normal brain tissue [161]. Malat1 (a long, noncoding RNA) activates autophagy and promotes cell proliferation by inhibiting miR-101, which downregulates the expression of autophagy-related genes such as *STMN1*, *RAB5A*, and *ATG4D* [164]. The levels of Malat1 were significantly increased in GBM biopsy samples with respect to adjacent normal tissue [164].

Various mechanisms have been described by which autophagy can induce carcinogenesis (Figure 2). Autophagy favors the survival of tumor cells in hypoxic areas (insufficient vascularization, limited supply of oxygen and nutrients) in solid tumors, such as glioblastoma, by degrading proteins, membranes, lipid droplets, and organelles to produce amino acids, fatty acids, and metabolic substrates that will allow tumor proliferation and survival. Furthermore, autophagy allows the maintenance of glioma stem-like cells, which induces therapeutic resistance and promotes tumor migration and invasion, and thus the recurrence of the disease [165].

Hypoxia (~3–0.1% oxygen) induces the activation of the hypoxia-inducible factor 1-alpha (HIF-1α), which promotes autophagy through the transcription regulation of autophagic genes, such as that coding for the Bcl-2/E1B 19-kDa-interacting protein (*BNIP3*), as well as *BNIP3L, BECN1*, and *ATG5* [165,166]. BNIP3/BNIP3L induces autophagy through the release of Beclin-1 from the Bcl-2/Beclin-1 or Bcl-xL/Beclin-1 complexes [98]. HIF-1α also induces angiogenesis, to ensure the availability of oxygen and nutrients for the survival of tumor cells, through the transcriptional regulation of VEGF [167]. Hypoxia levels and the expression of angiogenic factors are correlated with tumor grade and a poorer prognosis in patients with brain tumors [168]. A combination of chloroquine (autophagy inhibitor) and bevacizumab (anti-angiogenic, inhibitor of VEGF) exerted a potent antitumoral effect in mice xenotransplanted with glioma cells. It has been suggested that chloroquine inhibits autophagy caused by a hypoxic microenvironment; this effect is potentiated by bevacizumab, which induced resistance to the anti-angiogenic therapy [169]. On the other hand, silencing *Beclin-1* significantly decreased the expression of VEGF, MMP2, and HIF-1α in U87-MG glioma cells, leading to a reduction in the length of vasculogenic mimicry (VM) tubes under hypoxic conditions [170]. The formation of VM structures correlated with tumor grade in GBM patients, as well as with the expression of Beclin-1, VEGF, and MMP2 [170,171]. Hai-Bo et al. demonstrated that an increase in VM formation correlated with a poor prognosis and a higher expression of ATG5 and pKDR/VEGFR-2 in GBM patients. Autophagy can induce VM through the activation of pKDR/VEGFR by ROS generation and the ensuing activation of the PI3K-AKT pathway in glioma stem cells [172]. In addition, it has been reported that autophagy activation under hypoxia is related to transcriptional changes in genes, such as *BNIP3*, *BNIP3L*, *PIK3C3*, and *ATG9A.* Rahim et al. reported that inhibiting ATG9A decreased autophagy induction by hypoxia in intracranial glioma tumors, an effect linked to a significant decrease in tumor volume and an increase in mouse survival [173]. These results suggest that autophagy plays a major role in the aggressiveness and resistance of hypoxic regions of glioma, supporting the survival, proliferation, migration, and invasion of tumor cells. The tumor microenvironment promotes the growth of cancer cells through autophagy. Tumor cells generate ROS and promote autophagy in fibroblasts, which induce the glycolytic process resulting in high levels of pyruvate, lactate, and ketones; these products are used by tumor cells to cover their nutritional and energetic needs through the Krebs cycle/oxidative phosphorylation, due to their increased mitochondrial mass; apoptosis is inhibited, increasing tumor growth and migration/invasion [174,175,176]. Furthermore, oxidative stress in tumor cells induces the activation of pro-autophagy factors, such as LC3, BNIP3L, ATG16L, BNIP3, NF-κB, and HIF-1α, which promote the degradation of caveolin-1 (Cav-1), leading to autophagy activation. Cav-1 acts as a spontaneous suppressor of autophagy by binding (inactivating) the autophagic proteins ATG5, ATG12, the ATG12-ATG5 complex, and LC3B, as well as by modulating the expression of ATG16L, ATG5, ATG12, and LC3 [177]. A decreased expression of Cav-1 has been reported in brain endothelial tumor cells under hypoxic conditions [178]. Additionally, decreased Cav-1 levels correlated with higher levels of monocarboxylate transporters such as MCT4 and MCT1 [175,179]. MCT4, a promoter of the cell export of L-lactate and ketone bodies from glycolytic cells (tumor-associated fibroblasts), is regulated by HIF-1α under hypoxic conditions or when ROS levels increase [174,175]; then, MCT1 captures and uses this fuel for growth and tumor progression [180,181]. Hypoxia induces a positive regulation of MCT-1 [182], while a deletion of MCT-1 decreases glycolysis and metabolism rates in glioma cells through by regulating the mTOR signaling pathway, affecting tumor proliferation, migration, and invasion [183]. The expression of MCT1 in the plasmatic membrane is linked with HIF-1α and the carbonic anhydrase 9 (CAIX) in hypoxic areas of GBM tissues [184]. MCT1, MCT4, and their chaperone CD147 are overexpressed in biopsy samples from human GBM with respect to normal brain tissue, suggesting that MCT1 in the plasmatic membrane participates in the glycolytic phenotype of glioma, and that MCT4 in the cytosol may participate in the lactate-pyruvate mobilization to intracellular organelles [182]. A metabolic symbiosis has been observed between hypoxic and aerobic cells in glioblastoma. Treating human brain microvascular endothelial cells with conditioned medium (high in lactate) from U251 and SW1088 glioma cells cultured under hypoxic conditions induced a higher expression of MCT1, as well as an activation of oxidative metabolism and vessel assembly, which increased proliferation, migration, and angiogenic capacity of CT due to the assimilation of lactate from microenvironment via MCT-1. Furthermore, exposing endothelial cells to lactate induced the activation of AKT, mTOR, NF-кB, and HIF-1α, and of the lactate receptor GPR81. It was suggested that lactate (metabolic fuel) could act as a signaling molecule and participate in the brain endothelial cells-cancer cell crosstalk, and that MCT-1 inhibitors could be a promising therapeutic approach against GBM [185]. In addition, Duan et al. reported a decrease in glycolysis and an increase in oxidative phosphorylation in U251 glioma cells treated with lactic acid and deprived of glucose, as well as increased levels of MCT1, MCT4, and ATP by regulating the HIF-1α/C-MYC pathway [186]. Furthermore, those authors demonstrated that the expression of HIF-1, GLUT-1, LDH, and MCT4 increased in GBM tissue samples from the inner region of the tumor, whereas the expression of MCT1, C-MYC, and NRF1 is higher in the lateral section of the tumor. A higher expression of HIF-1, MCT1, and MCT4, along with the generation and transport of lactic acid in GBM indicates a poorer prognosis [186]. Those authors suggested that glioma cells in the inner region of the tumor are glycolytic, producing ATP and lactic acid and inducing the expression of MCT1 and MCT4. MCTT4 effluxes lactic acid from the inner region, and MCT1, located in the lateral region, takes up lactate, which is then catabolized by oxidative phosphorylation, which is up-regulated by CMYC and NRF1 [186].

Autophagy has been observed to facilitate tumor cell dissemination, favoring invasion and metastasis, by inhibiting anoikis. Anoikis is a form of programmed cellular death linked to detachment from the extracellular matrix (ECM) [187]. Under stress conditions, solid tumors, such as glioblastoma, exhibit resistance to cell death by inducing autophagy in cells detached from the primary tumor through PERK and the subsequent activation of the activating transcription factor 4 (ATF4), which induces the expression of autophagic genes, including *ATG5, ATG7*, and *ULK*, and of the antioxidant enzyme hemoxygenase-1, preventing anoikis and favoring the survival and migration of tumor cells [188]. AMPK was activated upon cell detachment from the ECM, either sustaining growth or preventing apoptosis [189]. Talukdar et al. demonstrated that autophagy contributes to the resistance to anoikis in glioma stem cells through the melanoma-differentiation associated protein 9 (MDA-9), which induces the hyperphosphorylation of Bcl-2 via PKC [190]. A higher expression of MDA-9 was linked to a higher glioma grade and a shorter patient survival [191].

On the other hand, autophagy induced by metabolic stress (lack of growth factors, nutrients, and/or oxygen) can extend the survival of apoptosis-deficient cancer cells, promoting a period of dormancy or quiescence and prompting cell proliferation once the stressing stimulus is over [192]. Cellular dormancy is characterized by a reversible arrest in the growth of single cells or cell groups as a response to stressing agents, such as hypoxia or cytotoxic drugs [193]; dormant cells can remain hidden and asymptomatic for a long time. Dormancy induction mediated by HIF-1α segregated by stem-like tumor cells in the peri-necrotic niche favors tumoral recurrence, decreasing the survival of GBM patients that received the standard therapy (surgery followed by radiation and chemotherapy [TMZ]) [193,194]. Magnus et al. found that glioma cell lines with low levels of tissue factor (TF) showed a dormant phenotype in vivo [112]. Those authors suggested that brain injury can activate TF, causing chronic coagulation and facilitating the recruitment of inflammatory cells that synthesize cytokines and oxidant products; these favor changes in dormant cells that allow them to express a malignant phenotype [112,195]. Additionally, a prolonged treatment with TMZ was proved to induce dormancy in glioma cell lines, which acquire characteristics of stem cells through the expression of dormancy markers, such as the insulin-like growth factor binding protein 5 (IGFBP5), the ephrin type-A receptor 5 (EphA5), and the histone cluster 1 H2b family member K (H2BK), and stem cell markers, such as the octamer binding transcription factor 4 (OCT4), sex determining region Y-box 2 (SOX2), and krüppel like factor 4 (KLF4) [196]. The signaling pathways that regulate tumor dormancy could be potential therapeutic targets to delay or stop glioblastoma recurrence after surgery [197].

## 4. Treatment Options for Glioblastoma

There is some debate on whether autophagy inhibition or induction could be exploited as a novel anti-cancer approach, and how autophagy-targeting drugs could be applied in the standard radio-chemotherapeutic regimens in cancer patients, since some data show that autophagy inhibition increases the efficacy of radiation and chemotherapeutic agents, increasing the cytotoxicity of various treatments [198,199,200]. On the other hand, an increase in autophagy enhances the therapeutic efficacy of several treatments by apoptosis induction [198,199,200]. Therefore, a treatment combining autophagy inducers and inhibitors could be a feasible strategy to improve the therapeutic effects of various agents currently in use.

### 4.1. The Standard Care: Temozolomide

The current treatment for glioblastoma and anaplastic astrocytoma consists of a surgical resection of the tumor, followed by radiotherapy (50–60 Gy) and an adjuvant chemotherapy treatment, mainly relying on TMZ [201]. TMZ is an orally administered, second-generation alkylating agent analogue to mitozolomide, with a good penetration into the central nervous system. It is transformed in the bloodstream into the active metabolite 3-methyl-(triazen-1-yl) imidazole-4-carboxamide, which donates a methyl group to some DNA bases (chiefly guanine), resulting in an erroneous matching of O^6^-methylguanine with thymidine during DNA replication; this leads to a halting in the G_2_/M phase of the cell cycle and subsequently to cell death [202]. Strupp et al. reported a beneficial effect of TMZ in recently diagnosed GBM patients. Treatment with TMZ plus radiation showed an increase in median survival from 12.1 months to 14.6 months, and an increase in 2-year survival rate from 10.4% to 26.5%. TMZ shows a tolerable toxicity level, with a 7% of grade III-IV hematological toxicity with concomitant radiation, and 14% when administered as an adjuvant [201]. This led the Food and Drug Administration (FDA) to approve TMZ for GBM treatment in 2005 [203].

While TMZ significantly increases median survival in GBM patients, its therapeutic effect is very modest, since cancer cells develop chemoresistance mechanisms. The therapeutic efficacy of alkylating agents in cancer has been reported as limited due to the presence of the repairing enzyme methylguanine-O^6^-methyltransferase (MGMT), which repairs DNA by directly removing the alkyl group from the genome of cells exposed to alkylating agents. Thus, treatment with alkylating agents is more effective if this protein is absent or if the *MGMT* gene promoter is methylated (as related with a loss of the gene) [204,205]. Helgi et al. found *MGMT* hypermethylation in 45% of GBM patients in the study by Strupp et al.; additionally, those authors found that methylation of the *MGMT* gene promoter is related with a significant increase in survival (21.7 vs. 15.3 months) and in progression-free survival (10.3 vs. 5.9 months) only in patients treated with TMZ plus radiotherapy [205].

To improve the efficacy of TMZ in GBM patients and lower the resistance of tumors to this drug, various works have searched for drugs that showed a synergistic effect with TMZ (Table 1). Li et al. found a high induction of cell death by autophagocytosis in U251 glioma cells treated with TMZ plus rapamycin (RAPA, a macrolide antibiotic with antitumor activity by inhibiting the mTOR pathway) even at low concentrations, thus preventing the toxic effects of TMZ on the patient. A combination of RAPA plus TMZ leads to an overexpression of Beclin-1 and LC3-II and an increased formation of acidic vesicular organelles (AVOs), considered as autophagy markers. These results suggest that autophagy in U251 cells is dependent of mTOR [206]. Jakubowicz-Gil et al. evaluated the efficacy of TMZ in combination with Sorafenib (a multikinase inhibitor with a high specificity for the protein Raf and a good permeability into the CNS) in GBM (T98G) and anaplastic astrocytoma (MOGGCCM) cells lines. Those results showed that Sorafenib acts synergistically with TMZ to trigger apoptosis, with a greater susceptibility in anaplastic astrocytoma cells than in GBM. An inhibition of the protein Raf reduced the activation of the Ras-Raf-MEK-ERK pathway, which induces cell death by apoptosis [207]. Another chemotherapeutic drug used in combination with TMZ is Momelotinib (MTB, also called CYT387); it is an aminopyrimidine derivative that inhibits the Janus kinases (JAK)-1/2. It was recently recognized that MTB promotes both chemosensitivity and a lower chemoresistance to TMZ in GBM cells. Co-treatment with MTB plus TMZ in GBM U251 cells and in a mouse xenograft model enhanced autophagy followed by apoptosis after inhibiting the phosphorylation of JAK2 and STAT3; furthermore, a decrease in the expression of Bcl-2 and Bcl-xL led to an inactivation of the signaling pathway JAK2/STAT3, resulting in apoptosis and autophagy. On the other hand, MTB decreased the levels of the protein MGMT, improving the effect of TMZ [208]. Pandey et al. studied the synergistic effect of TMZ with Roscovitine (RSV), a cyclin-dependent kinase (Cdk) inhibitor. Inhibiting Cdk5 in glioma cells (either in vivo or in vitro) increases the rate of autophagy and of caspase-3-dependent apoptosis. Co-treatment with RSV plus TMZ prevented the glioma growth, reduced angiogenesis, and limited tumor dissemination by reducing the number of reactive astrocytes [209].

Several attempts have been made to improve the efficacy of TZM to induce cell death in glioma cell lines, such as like C6. Recently, the research group led by Cheng and Zheng designed a TZM nanocarrier with 3-methyladenine (3-MA), a selective inhibitor of PI3K that blocks autophagosome formation. The nanoparticles were based on mesoporous silica a (MSNP) with polydopamine (PDA), an adhesion protein, coupled to a peptide with an Asn-Gly-Arg motif (NGR) that specifically recognizes the protein CD13, widely overexpressed in neoplastic tissue and angiogenic vasculature. The MSNP-TMZ-PDA-NGR drug significantly increased the rate of cell death by autophagy and apoptosis; adding 3-MA to the treatment inhibited autophagy, further increasing apoptosis and significantly improving the effect of TZM [210,211].

In the last few years, new approaches based on epigenetics and transcription regulation have been considered to enhance the therapeutic effect of TMZ in GBM. Ciechomska et al. reported that silencing the histone methyltransferase G9a (EHMT2 or G9a) with a specific competitive inhibitor, a diazepin-quinazolin-amine derivative (BIX01294), improved the efficacy of TMZ on the glioma cell lines LN18, U251, and LN18 GSCs. By inhibition G9a, BIX01294 reduced the histone H3 demethylation at lysine 9 (H3K9me2) and H3K27me2, favoring the upregulation of caspase-7 and PARP in LN18 cells. In contrast, BIX01294 slightly increased the levels of LC3-II in U251 cells. It is known that PTEN mutations can alter autophagy-activating pathways. When treated with BIX01294 plus TMZ, LN18 and U251 cells (both wild type and PTEN-mutant) showed reprogramming and sensitization to TMZ [212]. Yin et al. reported growth inhibition in U251 and LN229 glioma cell lines, and a higher toxicity of TZM when co-administered with Tubacin, a specific inhibitor of histone deacetylase 6 (HDAC6). Co-treatment with Tubacin increased the levels of LC3B-I, LC3B-II, and p62, indicating a phagosome accumulation [213]. By combining TMZ with a histone deacetylase inhibitor, suberoylanilide hydroxamic acid (SAHA), Gonçalves et al. found an exacerbated cytotoxicity, cell cycle arrested in the G_2_/M phase, and increased autophagy and apoptosis rates in both C6 and U251MG cells [214]. However, previous studies showed that GBM cells can be resistant to a prolonged stimulus with both drugs [215]. Therefore, those authors proposed to supplement the treatment with a late-autophagy inhibitor, 7-chloro-4-(4-diethylamino-1-methylbutylamino)-quinoline, chloroquine (CQ), a drug used in malaria. CQ, which is readily dissolved and absorbed, has an affinity for acidic cell compartments, such as lysosomes, endosomes, autophagosomes, and autophagolysosomes. CQ accumulation affects the function of lysosomes/autophagosomes, blocking autophagocytosis. A combined treatment with SAHA/TMZ/CQ further potentiated apoptosis, increasing the sensitivity of tumor cells to SAHA and TMZ [214]. The results reported by Gonçalves are consistent with those obtained by Huang, that inhibiting autophagy enhances the effect of therapeutic agents, such as TMZ. Considering that microRNAs control gene expression, Huang et al. identified that miR93 inhibits the expression of proteins, such as BECN1, ATG5, ATG4B, and SQSTM1, involved in autophagy. Treating GBM cells with TMZ and the ectopic expression of miR93 significantly inhibited autophagy, increased DNA damage and apoptosis induction by TMZ, leading to a significant reduction in tumor cell viability. By supplementing the combined administration of TMZ plus miR93 with another autophagy inhibitor, such as CQ or the ATG4B antagonist NSC185058 (NSC), tumor cell survival was further reduced [216]. Chen et al. found that the expression of miR-128 regulates multiple genes involved in cell death, survival, and cancer. The increased levels of miR-128 in U87-MG inhibited the mTOR pathway, enhancing the cytotoxicity of TMZ; they also produced ROS, with the ensuing loss of mitochondrial membrane potential (MMP), and promoted apoptosis and non-protective autophagy [217].

TMZ-resistant glioma cells were sensitized by blocking autophagy, following various strategies: (a) by overexpression of long non-coding RNAs if cancer susceptibility candidate 2 (CASC2); (b) by inhibiting the microRNA miR-193a-5p; (c) by inhibiting the mTOR signaling pathway; and (d) by the 3-MA autophagy inhibitor. All these interventions decreased tumor cell survival, migration, and invasion [218]. Li et al. also found that TMZ-resistant glioma cells (U87-MG and G131212) were sensitized by inducing the overexpression of miR-519a, which inhibits STAT3/Bcl-2, triggering autophagy-mediated apoptosis [208].

In another work, Zhang et al. found that glioma cells express miR-24-3p, whose function is to block mitochondrial degradation dependent on autophagy (mitophagy) from silencing by the Bcl-2/adenovirus E1B 19 kDa protein-interacting protein 3 (BNIP3). A downregulation of miR-24-3p by administering GSC (U-87 and U-251 cells) with a combined treatment of TMZ plus the endothelial-monocyte-activating polypeptide-II (EMAPII), a secretory polypeptide with anticancer properties, led to an overexpression of BNIP3, which induced mitophagy. TMZ plus EMAP-II showed a synergistical effect, increasing the sensitivity of tumor cells to TMZ, reducing GSC viability, migration, and invasion [219].

In a similar approach, GBM has been treated with the miRNA miR-224-3p. Hypoxia-induced autophagy leads to an overexpression of HIF-1α; in turn, it decreases the expression of miR-224-3p, whose target is ATG5, a protein essential for the formation of the autophagosome. ATG5 is involved in cell migration and the emergence of chemoresistance. Huang et al. observed that by overexpressing miR-224-3p, the expression of ATG5 was downregulated. This prevented the induction of autophagy by hypoxia, sensitizing the glioma to treatment with TMZ and decreasing cell migration [220].

The use of specific inhibitors of signaling pathways involved in tumorigenesis or in chemoresistance has also been explored. Chen et al. found that apoptosis and autophagy induced by treatment with TMZ is due in part to the blocking of the Notch3 signaling pathway induced by the overexpression of the cation transport regulator-like protein 1 (CHAC1); inducing the expression of CHAC1 in TMZ-treated glioma cells significantly decreased cell proliferation and migration, while the sensitivity to TMZ was increased. In addition to inhibiting Notch3, a combined treatment with TMZ plus CHAC1 stimulated ROS generation, increased intracellular Ca^2+^ levels, and led to the loss of MMP [221]. Another signaling pathway involved in cell survival, motility, and chemoresistance in GBM is the phosphoinositide 3-kinase (PI3K)-AKT signaling pathway. Shi et al. found that treating GBM cells with GDC-0941 (a highly specific inhibitor of PI3K) potentiates the sensitivity to TMZ. GDC-0941 arrested the cell cycle in the phase G_0_/G_1_, blocked the PI3K-AKT pathway, and increased the expression of GSK3β and p53, which trigger apoptosis and autophagy [222]. Alternatively, the combined use of intracellular Ca^2+^ and K^+^ mobilizing drugs plus an autophagy inhibitor (CQ or ATG5 deficiency), which led to cell death in glioma, has also been considered to prevent chemoresistance. The main ionophores used were the antibiotics Nigericin (isolated from *Streptomyces hygroscopicus*) and Salinomycin (from *Streptomyces albus*). These drugs increased the levels of intracellular ions and induced ROS generation, inhibiting autophagy and leading to cell death. A combination of TMZ with CQ or ATG5 deficiency plus Nigericin or Salinomycin synergistically suppressed spheroid formation in glioma cells. Guanosine (GUO, an endogenous nucleoside) shows modulating functions of glutamate release, exerting a neuroprotective effect. Oliveira et al. found that a combined treatment with TMZ, plus GUO acts synergistically, significantly decreasing growth and migration of glioma A172 cells, altering mitochondrial function, and increasing apoptosis [223].

The same research team treated A172 glioma cells with a combination of TMZ plus atorvastatin (ATOR), a synthetic statin capable of reducing the biosynthesis of cholesterol, regulating the abnormal lipid profile, and preventing cardiovascular events. ATOR has also shown anti-inflammatory effects, prevents revascularization; it crosses the blood-brain barrier (BBB) and acts as a neuroprotective agent, modulating glutamatergic transmission. ATOR has a cytotoxic effect, dependent of the activation of glutamatergic receptors, which induce apoptosis and autophagy by a mechanism yet unknown. While ATOR treatment decreased the viability, migration, and proliferation of glioma cells, it did not show a synergistic effect with TMZ, since the individual effect of these drugs is similar to the combined treatment [223]. Chu et al. reported that Thioridazine (THD, an antipsychotic drug with antineoplastic properties and the ability to cross the BBB) enhanced the action of TMZ on GBM U87-MG and GBM8401 cells. THD induced apoptosis and autophagy by increasing the activity of the AMP-activated protein kinase (AMPK) and stimulated the expression of the autophagocytosis regulatory proteins LC3 and P62. Chu et al. also reported that THD downregulated the Wnt/β-catenin signaling pathway, favoring the induction of autophagy-mediated apoptosis through a p62-dependent mechanism that involves caspase-8. THD also increased the activity of the proapoptotic protein Bax, reduced the expression of the antiapoptotic protein Bcl-xL, and induced apoptosis through a P53-independent pathway [224].

Elmaci et al. suggested that hormone therapy may act as an adjuvant to the standard treatment of glioma. Considering previous reports, they recognized that hormonal alterations in postmenopausal women are risk factors for a high-grade glioma. However, the likelihood of suffering from glioma decreased with the use of postmenopausal hormone replacement therapy (HRT). Elmaci et al. supplemented TMZ with Tibolone or medroxyprogesterone acetate (MPA), both progesterone analogs used in HRT. Both MPA and tibolone decreased spheroid growth in C6 glioma cells due to DNA synthesis suppression and the induction of mitophagy and autophagy [225]. Several research groups have studied the antineoplastic effects of phytopharmaceutical compounds to develop alternative treatments that supplement the current therapy. Polyphenols, terpenes, and cannabinoids have been evaluated as candidates to improve the efficacy of TMZ therapy.

β-Asarone (*cis*-2,4,5-trimethoxy-1-allyl phenyl) is a polyphenolic compound extracted from *Acorus tatarinowii* Schott and *Guatteria gaumeri* Greenman. Capable of crossing the BBB and be distributed into the CNS, it prevents tumor growth and induces cell death. Co-administering β-asarone plus TMZ inhibits cell growth in the U251 cell line, by arresting the cell cycle in the G_0_/G_1_ phase. β-Asarone improved the expression of Beclin-1 and expression of the protein P53, which promotes the induction of autophagy via P53/Bcl-2/Beclin-1 and P53/AMPK/mTOR in U251 cells [226]. A series of studies assayed the antineoplastic action of Honokiol (2-(4-hydroxy-3-prop-2-enyl-phenyl)-4-prop-2-enyl-phenol), a polyphenol extracted from *Magnolia officinalis*, capable of crossing the BBB and with a low toxicity. It induces autophagy by activating the signaling pathway p53/PI3K/Akt/mTOR; it also increases the levels of LC3-II, generates ROS, and increases the levels of vesicular acidic organelles [227,228,229]. Honokiol also produced DNA fragmentation, cell cycle arrest in the G_1_ phase, activation of caspase-3, and induction of p53/cyclinD1/CDK6/CDK4/E2F1-dependent apoptosis [227,228]. Recently, Chio et al. found that the combined treatment with TMZ plus Honokiol induced autophagy followed by apoptosis both in TMZ-sensitive (U87-MG and murine GL261) and TMZ-resistant (U87-MR-R9) glioma cells, improving the action of TMZ and sensitizing TMZ-resistant cells [229].

Recently, carnosic acid (CA), a polyphenolic compound derived from plants of the family Lamiaceae such as rosemary (*Rosmarinus officinalis*) or common sage (*Salvia officinalis*), was reported to show anticancer effects. Shao et al. found that the combined treatment of CA plus TMZ increased the antineoplastic and antimetastatic effects of TMZ, respectively. The synergistic action of CA and TMZ arrested the cell cycle in the G_0_/G_1_ phase and decreased MMP, leading to apoptosis [230]. Some apoptosis markers observed in TMZ-treated glioma cells are the segmentation of the enzyme poly (ADP-ribose) polymerase (PARP), whose function is to repair DNA; the drug also enhanced the segmentation of caspase-3 and reduced the levels of Cyclin B1. A combination of CA plus TMZ increased the presence of these apoptotic markers. In addition, a combination of both drugs significantly increased the levels of the protein LC3-II and decreased the level of p62, leading to increased autophagy rates. Moreover, CA inhibited the PI3K/AKT signaling pathway by decreasing directly AKT phosphorylation. Considering that the PI3K/AKT signaling pathway acts as a negative regulator of autophagy, CA prevents it from blocking autophagy [230].

Curcumin has also been used against GBM cells. It is a polyphenol isolated from the rhizome of *Curcuma longa*; its anti-proliferative and anti-angiogenic properties are derived from the alteration of the EGFR/PI3K/PTEN/RAS/STAT-3 signaling pathway [231]. Curcumin also decreases the expression of 43 (Cx43), a protein of the gap junction family with a key role in cell-to-cell communication. In GBM, the overexpression of Cx43 is related to gliomagenesis, cancer progression, and resistance to TMZ [232]. Huang et al. suggested TMZ-resistant GBM cells could be sensitized to the treatment by co-administering curcumin plus TMZ, downregulating Cx43. Curcumin sensitized TMZ-resistant cells, decreasing Cx43 expression by 40% via the ubiquitin-proteasome pathway, inducing high apoptosis rates [233].

Euphol, a tetracyclic triterpene alcohol widely known for its anti-inflammatory, analgesic, and antiviral effects, was first isolated from *Euphorbia tirucalli.* It causes cell death by an autophagy-dependent mechanism in neoplastic cells [234]. Silva et al. suggested that Euphol can be used in combination with TMZ to treat certain GBM cases. Euphol showed a heterogeneous cytotoxic effect in 14 GBM cell lines (21.4% were highly sensitive, 28.5% were moderately sensitive, and 50% were resistant); the highest sensitivity was observed in pediatric glioma. Despite having an antiapoptotic effect, the combined treatment of Euphol plus TMZ prevented cell proliferation and migration by inducing cell death by autophagy. However, autophagy inhibition by Bafilomycin A1 improved the cytotoxicity of the phytopharmaceutical, by a mechanism not studied yet [234].

Finally, the use of cannabinoids (extracted from *Cannabis sativa*) to enhance the antineoplastic effect of TMZ has shown promising results. Lopez-Valero et al. found that a combined administration of Δ9-tetrahydrocannabinol (THC) and cannabinol (CBD) plus TMZ decreased Ki67 levels, decreased cell proliferation, and significantly increased the expression of LC3 and DNA fragmentation in U87-MG glioma cells, indicative of cell death by autophagy and apoptosis [235].

### 4.2. Arsenic Trioxide

Arsenic trioxide (As_2_O_3_) is a drug approved by the FDA to treat acute promyelocytic leukemia [236]. It has been assayed in solid tumor cell lines, driving interest to extend its therapeutic application [237,238,239]. Its proapoptotic, antiproliferative, and antiangiogenic properties [240], which have led to its application in glioblastoma, have been widely studied (Table 2).

In a phase-I study including 17 patients (13 diagnosed with glioblastoma and 4 with anaplastic glioma), As_2_O_3_ was combined with TMZ and radiotherapy. The combination proved to be well tolerated and offered a good safety margin at a maximum tolerated TMZ dose of 75 mg/m^2^ + 0.2 mg/kg of As_2_O_3_. Adverse effects reported for this dose included elevated hepatic transaminases, prolonged QTc, thrombocytopenia, and neutropenia, similar to those observed in the treatment of acute promyelocytic leukemia. On the other hand, a 2-year follow-up was conducted to determine the mean survival time, being 19 months for glioblastoma patients (range: 5–57 months); four patients survived until the end of the study. The mean survival time for the patients who were followed-up until the end of the study was 45 months (range: 41–57 months) [241]. Another phase-I study including children diagnosed with infiltrative astrocytoma (21 patients in total, four of them with a diagnosis of glioblastoma) determined that As_2_O_3_ is safe and well tolerated at a dose of 0.15 mg/kg/day when administered along with radiotherapy, reporting nausea, vomit, fatigue, headache, and anorexia as adverse effects, similar to those observed when radiotherapy alone was administered. The survival range was 2 to 33 months, with a mean of 13 months for pediatric glioblastoma patients (range: 11–33 months) [242]. The same research group conducted a phase-II trial, administering the same doses to 42 patients with a mean age of 54 years (range: 24–80 years), 28 of whom were diagnosed with glioblastoma and 14 with anaplastic astrocytoma. The mean OS for glioblastoma patients was 17 months, and the median progression-free survival (PFS) was 7 months [243]. It has been suggested that As_2_O_3_ exerts its antineoplastic effects by acting on various signaling pathways.

As_2_O_3_ induces cell death through ROS production [244] by inhibiting the complex I of the respiratory chain (Figure 3) and antioxidant enzymes, such as glutathione peroxidase and thioredoxin reductase [245], as well as activating NADPH oxidase [246]. ROS induce apoptotic cell death by activating the c-jun N-terminal kinase (JNK) y and the proapoptotic protein BAX [247]. As_2_O_3_ has been reported to induce cyt c translocation from the mitochondrial matrix into the cytosol by activating BAX and negatively regulating the antiapoptotic protein Bcl-2 [248]. It has been demonstrated that changes in the BAX/Bcl-2 ratio induce changes in membrane potential, followed by the opening of permeability transition pores, with the ensuing release of cyt c into the cytosol, activation of caspases, and induction of mitochondrial apoptosis [247,248]. Sun et al. demonstrated that As_2_O_3_ induces apoptosis in C6 glioma cells by generating ROS, which promote the downregulation of the anti-apoptotic protein Bcl-2 and an up-regulation of the pro-apoptotic protein Bax [249]. In addition, As_2_O_3_ has been reported to induce apoptosis in U87 glioma cells through the p53 stability, the expression of Fas, FasL, and Bax, and the activation of caspase-3 and -9 [250]. On the other hand, As_2_O_3_ decreased the population of stem cell-like cancer cells (CSLC) in U87-MG, U251-MG, and U373MG glioma cell lines by blocking the Notch signaling pathway, decreasing the concentration of Nocht1 and Hes1 [251]. In addition, Linder et al. reported that a combination of As_2_O_3_ plus AT101 (a BH3-mimetics, inhibitor of the pro-apoptotic proteins Bcl-2, Bcl-xL, and Mcl-1) showed a synergistic effect against CSLC by inhibiting the Hedgehog and Notch signaling pathways [252]. In a neurosphere model of glioblastoma, a decrease in growth and proliferation was observed, as well as an increase in caspase-3 levels after treating with As_2_O_3_ the HSR-GBM1, 040622 (TMZ-resistant) and 040821 (TMZ-sensitive) cell lines. Additionally, an inhibition of PTCH1b, N-Myc, and GLI2 (transcriptional targets of the canonical Hedgehog pathway) was observed, as well as an inhibition of HES5 and HEY1 (transcriptional targets of the canonical Notch pathway) [253].

Another relevant drug target is the hERG1 channel, which is overexpressed in human glioblastoma samples and has been related to signals promoting cell proliferation and inhibiting apoptosis. Various studies have revealed that miR-133, which is overexpressed in astrocytes and acts as a negative regulator of hERG, is poorly expressed in the same cells. An inhibition of the expression of hERG has been reported by administering As_2_O_3_ to U251 glioma cells, probably due to a SRF-dependent upregulation of miR-133b, suggesting that miR-133b increases arsenic-induced apoptosis, taking the hERG channel as a target [254]. Other researchers have reported that the antineoplastic effects of As_2_O_3_ are due to the inhibition of the PI3K/AKT and NF-κΒ pathways [255,256].

NF-κΒ is a major pharmacological target, since it plays a key role in glioblastoma cell invasion and infiltration. It downregulates BAX and Bad, and it also promotes an overexpression of Bcl-2, Bfl-1, Bcl-w, and Survivin. In turn, the latter molecule stimulates the transcription and activity of telomerase, favoring cell survival. On the other hand, NF-κΒ can stimulate the urokinase-type plasminogen activator (uPA) through the lysosomal protease cathepsin B, inducing a metalloproteinase activity in the ECM, thus favoring cell migration and invasion. In a study on the U87-MG cell line, As_2_O_3_ significantly inhibited cell viability and proliferation, as well as the activity of cathepsin B, MMP2, MMP9, and telomerase. On the other hand, As_2_O_3_ increased cell adhesion, the activity of caspase 3, and cell death by apoptosis. These effects are mediated by a direct binding of As_2_O_3_ to IKK, which decreases the subsequent phosphorylation and degradation of IκBα, leading to a decrease in the activity of NF-κΒ [257].

The response to As_2_O_3_ varies from one tumor to another, and the involvement of some signaling pathways in As_2_O_3_-induced cell death by apoptosis has been challenged. In several tumors, caspase inhibition failed to prevent As_2_O_3_-induced cell death. Altogether, both caspase-dependent and -independent pathways have been reported as involved in cell-death mechanisms mediated by As_2_O_3_ [258,259,260]. The antiproliferative effects of As_2_O_3_ have been associated to a G_2_/M cell cycle arrest and autophagic cell death in human glioma cell lines, such as U87, T98G, A-172, U273, U251, and GB1, without evidence of apoptosis. However, a combined treatment with As_2_O_3_ plus the autophagy inhibitor Bafilomycin A1 increased the rate of cell death by apoptosis [257]. Kanazawa et al. reported that As_2_O_3_ induced mitochondrial damage and autophagy in U373-MG, T98G, and U87 cell lines by overexpression of the mitochondrial cell death protein BNIP3, suggesting that BNIP3 leads to cell death by autophagy, although the expression of BNPI3 in a glioma cell line induced autophagy even without As_3_O_2_ treatment [261]. As_2_O_3_ has also been reported to decrease cell viability, inducing cell cycle arrest in the G_2_/M stage, and apoptosis (by increasing the expression of BAX and decreasing Bcl-2) as well as autophagy (increasing the expression of Beclin-1, Atg5, LC3-II, and p62) in the U118 glioma cell line, by downregulating Survivin, through the inhibition of the PI3K/AKT and mTOR pathways, and the activation of MAPK mediators, such as ERK, JNK, and p38 [262]. Survivin is a member of the apoptosis inhibitor protein family (IAP) that inhibits caspases and promotes resistance to several therapeutic agents [263]. On the other hand, Li et al. reported that the combination of an autophagy inhibitor (Chloroquine) plus an autophagy inductor like As_2_O_3_ induced both autophagy and apoptosis, with accumulation of autophagic vacuoles by the action of Beclin-1 [264]. The growth in autophagosome can produce ROS, leading to an increase in mitochondrial permeability and the subsequent release of cyt c into the cytosol, triggering the activity of caspase-3 [72]. These authors proposed that a combination of negative regulators of the autophagic process with an apoptosis inducer could be a feasible solution to improve the efficacy of glioblastoma treatment [264].

### 4.3. Chloroquine

Chloroquine (CQ) a synthetic 4-aminoquinoline, was first used as an antimalarial agent [265]. It is a weak base, and at the physiological pH in the bloodstream, it is in a deprotonated state, easily diffusing through cell membranes. Upon entering the cell, CQ is protonated and trapped in acidic organelles, such as lysosomes, which maintain an inner pH close to 5.0 through the activity of lysosomal H^+^-ATPases. As these proteins pump H^+^ ions into the acidic vesicles, more CQ diffuses from the cytoplasm into the lysosome following a pH gradient, causing an irreversible CQ accumulation, and increasing pH by trapping H^+^ ions [266]. In glioblastoma cells, this process prevents the autophagosome-lysosome fusion, inhibiting the final stage of autophagy and inducing cell death by apoptosis [267] (Figure 3). However, this mechanism fails to explain the observed cell damage, apparently specific for tumor cells.

The interest in treating glioblastoma patients with CQ has grown in recent years (Table 2). In a retrospective analysis, patients diagnosed with glioblastoma, either EGFRvIII-positive or not, were treated with CQ in combination with carmustine, radiotherapy, and surgery; CQ administration improved the median survival in patients with EGFRvIII-negative GBM from 5 to 10 months, while it increased from 3 to 15 months in patients with EGFRvIII-positive GBM [268]. The authors suggested a potential use of chloroquine in EGFRvIII-positive tumors [268]. It has been reported that EGFRvIII-expressing glioma cells underwent autophagy, which allowed them to be more resistant to several therapies and to metabolic stressors, such as hypoxia and starvation [268]. By inhibiting autophagy, CQ reduced hypoxia and accelerated the growth of EGFRvIII-positive tumor xenografts [268]. In addition, CQ was used in a phase-I trial along with TMZ, radiotherapy, and surgery, the conventional treatment for glioblastoma. Patients were administered with CQ over the observation period, and a longer survival time was observed in CQ-treated patients (35 ± 5 months) with respect to the control group (11 ± 2 months) [269]. Then, a randomized, double-blind trial was conducted comparing CQ with placebo administered as an adjuvant to the conventional treatment and setting a 12-month follow-up period. The maximum and minimum follow-up time was 59 and 5 months, respectively. Mean survival post-surgery was 24 months for CQ-treated patients and 11 months for controls [270]. A third trial, where CQ was administered to 41 patients and the outcome was compared with 82 controls who only received the conventional treatment, demonstrated once again an increase in survival time for those patients whose treatment was supplemented with CQ (25 ± 3.4 months) with respect to the control group (11.4 ± 1.3 months) [271]. It is noteworthy that low CQ doses (150 mg per day) were used in all three studies, so the drug was well tolerated, showing a non-significant decrease in leukocyte and platelet counts with respect to the control group [270]. The mechanisms through which CQ increases TMZ cytotoxicity are unknown. However, Golden et al. showed that CQ synergizes the cytotoxic effect of TMZ on U251, LN229, and U87 glioma cell lines, as well as on U251-TMZ^R^ (TMZ-resistant) by halting PI3K-(III)-Beclin-1 and GRP78 (endoplasmic reticulum (ER) chaperone)-dependent autophagy, as well as by inducing the conversion of LC3-I into LC3-II; furthermore, it increases the expression of proapoptotic proteins, such as the CCAT enhancer-binding protein (C-EBP) homologous protein (CHOP/GADD) and PARP. Glioblastoma cells overexpress and induce the production of the survival proteins GRP78/BiP, which decrease ER stress-associated apoptosis and are components of the autophagic pathway. These proteins limit the cytotoxic effect of CQ and TMZ, so basal tumor GRP78 levels can be predictors of sensitivity to CQ and resistance to TMZ [272]. Tumor cells with chronic ER stress resulting from misfolded proteins undergo autophagy through the PI3K(III)-Beclin-1 and UPR pathways. UPR leads to the activation of GRP78, which sustains ER integrity and promotes autophagosome formation and the subsequent fusion with lysosomes. Then, autophagolysosomes degrade misfolded proteins and reduce ER stress, while providing nutrients required to synthesize proteins, nucleic acids, lipids, and carbohydrates. Hori et al. also demonstrated that CQ potentiates the antineoplastic effect of TMZ on glioma cell lines by ROS generation and by inhibiting mitochondrial autophagy. Additionally, it was demonstrated that a loss of Beclin-1 activity, involved in phagosome formation, increases the amount of mitochondrial ROS [273]. Furthermore, a synergistic effect of TMZ and CQ was reported on the U87 (wild type p53) cell line, whereas CQ increased the apoptosis induced by TMZ via p53, which is dependent of autophagy. However, the CQ-TMZ combined treatment inhibited cell proliferation in the U373 (mutant p53) cell line by arresting the cell cycle in the G_2_/M phase. Those authors proposed that the antineoplastic response to the CQ-TMZ combination depends on the status of p53 [267].

On the other hand, Geng et al. reported that CQ induced cell death in five glioma cell lines by a p53-independent mechanism. A redistribution of the cathepsin D lysosomal protease was also induced, along with an accumulation of lysosomal vacuoles, suggesting that alterations in lysosomal function play an important role in CQ-induced cell death [273]. Ionizing radiation has been reported to increase autophagy in human glioblastoma cells, thus leading to radioresistance. A synergic antineoplastic effect was observed when CQ was used in combination with radiotherapy on U87 glioma initiating cells (U87 GICs) with respect to either radiation or CQ treatment alone, through apoptosis induction and autophagy inhibition, promoting cell cycle arrest in the G_0_/G_1_ phase, increasing the expression of caspase-3, and decreasing the expression of Bcl-2 and autophagosome formation [267]. Firat et al. demonstrated that a triple combination of low CQ doses (5–30 μM) plus γ-irradiation and an inhibitor of the PI3K/AKT/mTOR pathway, such as LY294002, the AKT inhibitor III, or PI-103, has a strong antineoplastic effect on radioresistant stem-like tumor cells isolated from human glioma biopsies, higher than that of a double combination of γ-irradiation and CQ (10–50 μM), through apoptosis induction and autophagy inhibition [274]. Furthermore, Sun et al. reported that a CQ concentration of 50 μΜ significantly decreased autophagy rates in F98 glioma cells that stably expressed CD133, inhibiting cell resistance to nutrient deprivation, inducing apoptosis and necrosis. The authors suggested that CD133 promoted autophagy by taking part in autophagosome formation and autophagosomal degradation, thus promoting cell survival in a nutrient-lacking tumor microenvironment [275]. CD133 has also been reported to aid glucose intake and ATP synthesis in tumor cells [276]. Additionally, it has been demonstrated that autophagy inhibition by CQ increases the anti-proliferative and anti-invasive effects of sorafenib on U373 and LN229 glioma cells [277]. Gonçalves et al. reported that CQ co-administration enhanced the antineoplastic effect of both vorinostat (an inhibitor of histone deacetylases) and TMZ alone on orthotopic GL261 glioma [214].

### 4.4. Metformin

Metformin, a drug of the biguanide group, was discovered in the 1920s while studying isoamyl guanidine isolated from the extract of *Galega officinalis*. It is used since 1957 in Europe to treat diabetic patients due to its capacity to decrease blood glucose levels and increase the sensibility to insulin. Metformin inhibits gluconeogenesis, increases the recapture of glucose from muscle, and increases the levels of the glucagon-like peptide 1 (GLP-1) [278]. Diabetic patients were observed to show a lower risk to develop cancer in a following-up period of 8 years, gaining interest for its potential as an antineoplastic drug. This effect could be due to the activation of AMPK, an enzyme regulated by the hepatic kinase B1 (LKB1), a well-known tumoral suppressor [279] (Figure 3).

It has been proved that glioma cell lines, such as U87, LN18, U251, and SF767 are sensitive to metformin treatment (Table 2). In fact, metformin reduced cell proliferation by downregulating the AKT pathway; this effect is more marked in LN18 and SF767 cell lines, which have a functional PTEN (wild type). Furthermore, metformin induces cell cycle arrest, blocking the progression in the G_0_/G_1_ phase, as well as cell death associated with JNK activation, inhibition of complex I from the respiratory chain at a mitochondrial level, activation of AMPK, and inhibition of mTOR, inducing oxidative stress. Interestingly, metformin is effective in GBM cells resistant to the standard treatment. For example, the cell line SF767 has shown resistance to standard treatment due to a non-methylated MGMT (O-6-methylguanine-DNA-methyltransferase); metformin increased the sensibility of SF767 cells to TMZ and radiotherapy [280]. In addition, metformin decreased cell adhesion and invasion rates in the U251 line by inhibiting the expression of fibulin-3 and MMP2 [281]. Another possible mechanism underlying the action of metformin on glioma cell lines is a decrease in STAT3 phosphorylation. When a STAT3 inhibitor is combined with metformin, tumor proliferation decreases significantly. In contrast, phosphorylated STAT3 results in more aggressive tumor cells, worsening patient survival [282].

Other studies have reported an increased SOX2 expression in TMZ-resistant glioblastoma cells; however, treatment with metformin decreased SOX2 expression, and therefore proliferation capacity in those cells [283]. In addition, the combined administration of metformin plus sorafenib exerted anti-proliferative and pro-apoptotic effects on glioma stem cells through ROS generation, lipoperoxidation, and inhibition of efflux pumps [284]. Kim et al. reported that metformin downregulated the levels of P-glycoprotein transcript and protein by transcriptional inhibition of NF-κB and CREB via AMPK [285]. Furthermore, Sesen et al. demonstrated that metformin induced autophagy and apoptosis in glioma cells by activating AMPK and Redd1 and inhibiting mTOR [280]. A combination of As_2_O_3_ plus metformin promoted autophagy and apoptosis in glioma cells through the activation of AMPK-FOXO3 [286,287].

However, retrospective studies on the effect of metformin on glioblastoma patients reported controversial results. In a case-control study, Seliger et al. observed that the risk for glioma was lower in patients with chronic diabetes; still, no correlation was found between patients using metformin and a reduction in glioma risk (OR = 0.72; CI95% = 0.38–1.39) [288]. Adeberg et al. evaluated the survival of diabetic patients diagnosed with glioblastoma; the patients received surgery and TMZ. PFS in patients who were not administered with metformin treatment was 6.77 months (range: 1–54 months), with an OS of 14.58 months (range: 1–84 months). Metformin-treated patients showed a significantly better PFS (10.13 months, *p* = 0.018). OS was lower in patients who received corticosteroid therapy and those who had hyperglycemia [289]. In another retrospective cohort study, the survival of patients with high-grade glioma and either with or without metformin treatment was analyzed. Patients with grade-III glioma who received metformin had a better OS (HR = 0.30; CI95% = 0.11–0.81) and PFS (HR = 0.29; CI95% = 0.11–0.78) with respect to grade-IV glioma patients. This could be due to an IDH mutation, frequent in patients with low-grade glioma. In fact, this mutation causes a deficiency in the reductive anaplerosis of glutamate; this effect, accentuated by metformin treatment, is lethal to the cell [290]. In another study, the association between metformin treatment at the time of glioma diagnosis and during the concomitant treatment with TMZ and radiotherapy was evaluated. Metformin treatment did not significantly improved OS nor PFS [291]. To date, no randomized clinical trials have provided more reliable evidence on the use of metformin as complementary treatment in glioma.

### 4.5. Small-Molecule Inhibitors

The aggressiveness of glioblastoma is due to a highly mutated genome (amplification and overexpression of EGFR, PDGFR, and VEGFR, and deletion or inactivation of PTEN, RB, P53, p14^ARF^ and p16^INK4A^) [292], which favors a hyperactivation of signaling pathways (RAS/RAF/MEK/ERK and PI3K/AKT/mTOR) that regulate cell proliferation, angiogenesis, migration, and invasion, as well as cell-death mechanisms. Thus, most therapeutic strategies against glioblastoma are aimed to downregulate these signaling pathways, inducing cell death by apoptosis and autophagy. By inactivating cell targets that inhibit both death processes, the sensitivity of tumor cells to various therapeutic agents is increased. Thus, the use of tyrosine kinase inhibitors (TKI) could be an effective approach for patients with tumors affecting the nervous central system, such as glioblastoma. TKIs are low-weight molecules that generally bind tyrosine kinase domains in intracellular receptors (EGFR, PDGFR, and VEGFR), competitively blocking the binding to ATP, and therefore preventing receptor auto-phosphorylation and the activation of intracellular signaling cascades (RAS/RAF/MAPK, PI3K/PTEN/AKT/mTOR, and PKC). The TKIs most used to treat GBM in the clinical practice are gefitinib (Iressa^®^, Astra-Zeneca) and erlotinib (Tarceva^®^, OSI 774), being both first-generation EGFR inhibitors; dacomitinib, a second-generation Pan-HER inhibitor (PF299804, Pfizer); imatinib mesylate, a PDGFR inhibitor; and VEGFR inhibitors such as sunitinib and vandetanib (Table 2 and Figure 3).

#### 4.5.1. Erlotinib

In vitro studies have shown that erlotinib inhibits EGFR-dependent cell proliferation at sub-micromolar concentrations and induces arrest in the G_1_ phase of the cell cycle, without inducing apoptosis [293]. Erlotinib inhibited apoptotic cell death by increasing the levels of the heat-hock antiapoptotic protein αβ-crystallin, which inhibits the activation of caspase-3. Meanwhile, high doses of erlotinib (much higher than the therapeutic dose) cause cell death by activating caspase-independent pathways and the autophagic pathway. This toxic threshold can be raised by a combined administration of erlotinib with the autophagy inhibitor CQ, causing the cell survival programming to be shifted to an apoptotic cell death programming [294]. Those authors suggested that using erlotinib plus autophagy inhibitors on apoptosis-resistant glioma cells increases the antineoplastic effect of the TKI [294].

A combination of erlotinib and sorafenib (VEGFR2, PDGFR, and RAF inhibitor) significantly increased cell death levels in GICs with respect to erlotinib alone, both through apoptosis induction (accumulation of the proapoptotic protein BIM, downregulation of the antiapoptotic proteins Bcl-2 and Bcl-xL, and PARP cleaving) autophagy (accumulation of LC3-II and phosphorylation of mTOR) [295]. A significant decline of pAkt, S6, and ERK, as well as decreased nuclear levels of PKM2 and β-catenin were observed, suggesting that inhibiting the RTKs and RAF/MEK pathways synergize the antineoplastic effect of erlotinib [295]. The MEK/ERK/NF-κB pathway induces a nuclear translocation of PKM2, which binds to β-catenin; this allows the transcription of important genes for tumor cell proliferation, angiogenesis, and invasion [296,297]. Karpel-Massler demonstrated that administering a combination of erlotinib plus NSC23766 (a RAC1 inhibitor) to A173, T98, and U87 human glioma cell lines inhibited cell proliferation by inducing autophagy and caspase-3-independent apoptosis, with a reduced expression of the protein Survivin [298]. Notwithstanding the encouraging antineoplastic effect of erlotinib in preclinical studies, it has shown a poor activity in clinical trials. In phase-I/II studies, recurring GBM patients were treated with erlotinib, carmustine, or TMZ; the 6-month progression-free survival rate (PFS-6) was 11.4% in the erlotinib group and 24% in the carmustine or TMZ groups [299]. In another trial, co-treatment with erlotinib plus TMZ, either before or after radiation therapy, led to an increase in PFS-6 from 14.1 months (historical control) to 19.1 months [300]. A phase-II trial exploring the possible benefits of erlotinib combined with bevacizumab in recently diagnosed GBM patients who showed a non-methylated *MGMT* promoter showed no statistically significant differences in patient survival (mean survival was 13.2 months, compared with 12.7 months reported in a historical control). No significant differences were observed neither when data were stratified by age or clinical improvement [301]. Nevertheless, another research group used the same therapeutic combination and reported a discrete improvement in the mean PFS (13.5 months vs. the historical mean of 8.6 months), but not in mean survival time (19.8 months vs. the historical mean of 18 months). Furthermore, no correlation was found between the expression of the protein PTEN, the amplification of EGFR, nor the methylation of the *MGMT* promoter and any improvement in patient survival [302]. A study assaying a combined treatment with erlotinib plus TMZ and radiotherapy in pediatric patients (median age, 10 years; range: 3–19 years) diagnosed with high-grade glioma, showed again that the treatment failed to change the prognosis of the disease, giving a mean PFS of 10.7 and 6.3 months, and a mean survival of 15.4 and 11.7 months in astrocytoma anaplastic and GBM, respectively [303]. No statistically significant differences in patient survival were found in a phase-II trial where adult patients suffering from recurrent high-grade glioma were administered with erlotinib plus temsirolimus. The presence of the EGFRvIII mutation, EGFR amplification, and PTEN expression were correlated with patient survival, with no successful results [304].

#### 4.5.2. Gefitinib

Gefitinib has been reported to bind ERBB-3, inhibiting the EGFR cell signaling, which induces arrest in the G_1_ phase of the cell cycle and inactivates the ERK and AKT kinases [305]. Gefitinib has also been reported to inhibit cellular migration in vivo [306]. In addition, Chang et al. showed that gefitinib induced apoptosis in several glioblastoma cell lines by dephosphorylating the proapoptotic protein Bad, with the ensuing translocation of BAX to mitochondria and the activation of caspase-9 [307]. This same research group demonstrated that gefitinib inhibits cell proliferation in glioma cells by inducing autophagy. In various in vitro and in vivo models, treatment with gefitinib at low doses (5–20 μM) induced autophagy by activating the LKB1/AMPK signaling pathway, inhibited cell proliferation in U87, T98G, and H4 glioma cells, and reduced tumor volume in xenotransplanted mice. High gefitinib doses, on the other hand, led to cell death by EGFR inhibition-independent apoptosis [308]. A synergic effect has been observed in the combined treatment with gefitinib plus MK-2206 (an AKT inhibitor) in LN229 and T98G glioma cells and in naked mice with tumor xenograft; in both models, apoptotic (Survivin downregulation and Bim upregulation) and autophagic (LC3-II elevation) processes were activated by inhibiting the phosphorylation of a key component of the AKT/mTOR/S6K signaling pathway. A shift from autophagy to apoptosis was observed in prolonged treatments [309]. A synergistic effect of the combined treatment with gefitinib and valproic acid on glioma cells has been reported; the combination acts by autophagy induction, ROS generation, and LKB1/AMPK and ULK1 activation [310]. The authors suggested that gefitinib and valproic acid induce ROS formation by activating NADPH oxidase [310]. A combined treatment with β-elemene (a secondary metabolite from *Curcuma wenyujin*) and gefitinib induced apoptosis and autophagy in glioma cell lines by inhibiting the EGFR signaling. Those authors suggested that β-elemene can reduce resistance to gefitinib in glioma cell lines and acts as an adjuvant, increasing the efficacy of the EGFR inhibitor [311].

The promising results in preclinical assays have not been successfully extrapolated to the clinic. Phase-I studies have determined a maximum tolerated dose of 500 mg (750 mg in patients receiving enzymatic inducers, such as anticonvulsants). Some commonly reported adverse effects are rash, diarrhea, and fatigue [312,313]. In a phase-I/II trial, 12.7% of gefitinib-treated glioblastoma patients showed partial tumor regression; however, OS was not better than that of historical controls. Chakravarti et al. have reported that a combined treatment with gefitinib and radiation in patients recently diagnosed with glioblastoma exhibited a PFS-6 of 40% and an OS of 11.5 months, not significantly different from radiotherapy-only historical control (11.0 months). Stratified data by age showed a better clinical response in younger patients [313]. According to other authors, no significant changes in patient survival were observed when gefitinib was administered as an adjuvant for radiotherapy [314,315].

#### 4.5.3. Dacomitinib

Dacomitinib is a second generation, orally administered, irreversible inhibitor of HER1, HER2, and HER4. In preclinical studies, it induced apoptosis in C6 glioma cells in vivo [316]. Dacomitinib also inhibited cell proliferation and decreased the activation (phosphorylation) levels of EGFR, promoting the inactivation of Akt and ERK1/2 in U87-MG (PTEN-negative) cells, either with or without EGFRvIII expression. In U87 xenografts, dacomitinib induced a modest increase in OS [317]. Zahonero et al. demonstrated that dacomitinib inhibited the EGFR signaling pathway in EGFR-amplificated GBM xenografts, decreasing cell proliferation and tumor volume by inactivating AKT, ERK, and S6 and promoting apoptosis [318]. Those authors also demonstrated that dacomitinib can cross the BBB, decreasing intracranial tumor growth rates by inhibiting the EGFR signaling and the self-renewal of carcinogenic stem cells [318]. In addition, dacomitinib has shown antineoplastic effects independently of the expression of mutated isoforms of EGFR, such as EGFRvIII and EGFRvII [319,320]. However, Zhu et al. reported that glioma cells carrying the EGFRvIII mutation show an overactivation of mTOR, being less responsive to the anti-tumor effect of dacomitinib [319]; the co-treatment of PF-05212384 (PI3K/mTOR inhibitor) re-sensitizes the cells to the TKI. Dacomitinib significantly decreased cell viability and induced apoptosis by inactivating the EGFR/PI3K/MAPK/mTOR pathway in glioma (PTEN mutant) cells with respect to the treatment with PF-05212384 or dacomitinib alone [319].

Dacomitinib exhibited a poor antineoplastic activity in a multicenter, open label, phase-II clinical trial on two cohorts of patients with recurrent glioblastoma that showed EGFR amplification, either with or without EGFRvIII [320]. In both cohorts, PFS-6 was 10.6% (median, 2.6 months), and the drug had a toxicity profile that included rash and diarrhea [320]. Preclinical data suggested that tumor heterogeneity, loss of PTEN, the BBB, and the difficulty to achieve an adequate drug concentration hindered the effectiveness of dacomitinib [320,321]. It has been suggested that EGFR suppresses autophagy via Beclin-1 [322]; it is possible that autophagy suppression further contributed to tumor progression and resistance to dacomitinib in glioma patients [319,320].

#### 4.5.4. Imatinib

Imatinib inhibits some receptors with tyrosine kinase activity, such as PDGFRα and PDGFRβ, as well as non-receptor kinase proteins (C-abl and BRCP) [323]. Imatinib induces autophagy (formation of autophagic vacuoles and LC3-II) in human glioma cells, by increasing the phosphorylation of ERK1/2 and inhibiting the AKT/mTOR signaling pathway. Early autophagic inhibition with 3-methyladenine (inhibitor of class-III PI3K) reduces the antineoplastic effect of imatinib, while late autophagy inhibition by Bafilomycin A1 (an H1-ATPase inhibitor) increases the toxicity of imatinib and favors apoptosis. It was suggested that an appropriate modulation of the autophagic process could be an effective therapy against cancer cells [324]. Co-treatment with imatinib plus the antidepressant chloropyramine enhanced the anti-neoplastic activity of imatinib on monolayer and spheroid cultures of rat glioma cells by autophagy activation followed by apoptosis. However, treatment with imatinib alone, led to chemoresistance due to autophagy induction [325]. The authors suggested that chloropyramine could facilitate the cellular penetration of imatinib by inhibiting the P-glycoprotein (P-gp) and the respiratory complex III, generating ROS and apoptosis [325,326].

A synergic antitumoral effect was observed in C6 cells and spheroids co-treated with imatinib plus Carvedilol (a cardiovascular drug) resulting from autophagy induction, cell cycle arrest in the G_0_/G_1_ phase, and apoptotic cell death due to ROS generation and mitochondrial damage [327]. It was suggested that the antineoplastic effect of imatinib on glioma cells could be increased by suppressing autophagy, and a finer modulation of this process may sensitize tumor cells to anticancer treatments [327]. However, the combination of imatinib and nilotinib increased cell migration and invasion in glioma stem cells through the activation of p130Cas, the focal adhesion kinase (FAK), and paxillin, without affecting the activity of PDGFRβ and c-Abl [328]. Nilotinib, a second-generation PDGFRα inhibitor, has shown anti-proliferative effects on SJ-G2 glioma cells of pediatric origin, by inhibiting the ERK and AKT signaling pathways [329]. Imatinib has shown a poor antitumor effect in glioblastoma patients, with a PFS-6 less than 16% and a radiological response below 6% [330]; similarly, Wen et al. reported a PFS-6 of 10% [331]. It has been suggested that the low response to imatinib treatment in high-grade glioblastoma could be due to a low drug availability at the tumor mass. However, co-treatment with hydroxyurea plus imatinib in GBM recurrent patients has shown promising results, with a PFS-6 of 27% and a median PFS of 3.5 months, with a 42% of radiological response. The authors suggested that imatinib favored the transport of hydroxyurea into the CNS by inhibiting P-gp-mediated efflux [332].

#### 4.5.5. Sunitinib

Sunitinib is an inhibitor of angiogenesis-inducing receptors, such as VEGFR, PDGFRα/β, the colony-stimulating factor-1 receptor (CSF1R), the Fms-like tyrosine kinase-3 receptor (FLT3), and the stem cell-factor receptor (Kit). Angiogenesis has been proved to favor tumoral promotion, migration, invasion, resistance to therapy and apoptosis, and relapses [333]. Sunitinib reduces cell proliferation by promoting an arrest in the G_2_/M phase of cell cycle on GL15 glioma cells; it also induces apoptosis by activating caspase-3, and shows an anti-invasive effect mediated by a decrease in the activation of Src and FAK [334]. Sunitinib decreases the accumulation of extravasated vascular accessory cells (circulating bone marrow-derived myeloid cells) in the perivascular niche of glioma [335]. Grunewald et al. reported that circulating bone marrow-derived myeloid cells accumulate in the perivascular niche, leading to angiogenesis via VEGF/small chemokine stromal-derived factor-1 (SDF1) [336]. Sunitinib induced autophagy in several cancer cell types by promoting the expression of Beclin, the conversion of LC3-I to LC3-II, and the formation of autophagosomes [337]. The co-administration of sunitinib with other drugs to potentiate its anti-neoplastic effects has been explored. For instance, CQ increases the cytotoxic and antiangiogenic capacity of sunitinib by increasing CD34 levels, resulting in decreased rates of blood vessel formation and apoptosis through the inhibition of Survivin and the activation of caspases [338]. Li et al. reported that a combination of sunitinib plus CQ induced apoptosis by increasing the levels of Bcl-2 and p53 and decreasing Bax expression [339]. On the other hand, a combination of acetylsalicylic acid plus sunitinib showed higher anti-proliferative, anti-angiogenic, anti-invasive, and pro-apoptotic effects on human primary GBM-endothelial cells than either drug alone, by inhibiting the release of VEGF, VEGFR1/2, HRAS, PI3K, AKT, MEK, and ERK, as well as by down-regulating Bcl-2 and up-regulating Bax [338,340]. A combination of sunitinib plus gefitinib or sunitinib and sorafenib inhibited neurosphere proliferation and growth by inhibiting AKT, MAPK, and STAT3 [341]. miR-145 Mimic enhanced the anti-neoplastic effect of sunitinib on U87 Cells by downregulating Cyclin D; it also increased the activity of P-gp and the breast cancer resistance protein (Bcrp) [342]. Sunitinib was also reported to improve the antiproliferative and proapoptotic effects of TMZ in vivo; however, this combination induced vascular resistance by increasing the expression of Ang-1 and Tie 2 [343]. Ang-1 binds Tie2 to promote the maturation and remodeling of the tumor vascular network [344]. EphB4 favored the vascular resistance to sunitinib by disturbing vascular morphogenesis, pericyte coverage, and inhibiting apoptosis in vitro and in vivo [345]. Chen et al. reported that EphB4 induced the activation of EGFR [346]. It was reported that the G-protein-coupled receptor (CXCR4) and the small chemokine stromal-derived factor-1α (SDF1α) are overexpressed in glioma cells xenografted in animals treated with bevacizumab and sunitinib, increasing the resistance to therapy [347]. Gravina et al. reported that PRX177561 (a CXCR4 inhibitor) increased the anti-angiogenic effect of sunitinib and bevacizumab in intracranial GBM-xenografted nude mice, reducing tumoral growth and blood vessel density, while improving the survival time of the animals by inducing apoptosis and palisading necrosis [348].

Clinical studies on sunitinib-treated patients with recurrent primary glioma failed to show an objective radiological response, with a PFS-6 of 12.5%, median PFS of 2.2 months, and median OS of 9.2 months [349]. A low antitumor response to sunitinib and irinotecan and lomustine has been reported in recurrent glioblastoma patients [350]. Oberoi et al. suggested that the poor response to sunitinib in CNS tumors is due to a low penetration of the drug into the brain, due to the presence of P-gp and Bcrp.

#### 4.5.6. Sorafenib

Sorafenib is a TKI that reduces tumor cell viability, angiogenesis, migration, and invasion; it induces apoptosis in tumor cells by inhibiting the activity of VEGFR2/3, PDGFRβ, FLT3, and c-KIT [351]. In addition, sorafenib inhibits the RAF/MEK/ERK pathway and downregulates the anti-apoptotic protein Mcl-1 [352,353,354]. Sorafenib reduced the activity of VEGFR2 in glioma microvascular endothelial cells [355]. Sorafenib inhibited cell proliferation and promoted apoptosis in glioma initiating cells, by downregulating the activity of PI3K/Akt, MAPK, and Mcl-1 [356]. On the other hand, sorafenib inhibited STAT3 phosphorylation and decreased the levels of the proteins cyclin D, cyclin E, and Mcl-1 in glioma cells [357]. Kiprianova et al. demonstrated that sorafenib increased the efficacy of ABT-737 (a BH3 mimetic: Bcl-2, Bcl-xL, and Bcl-W inhibitor) for apoptosis induction on glioma cells by inhibiting STAT3 phosphorylation, which induced the expression of Mcl-1 [358]. Furthermore, sorafenib has been described to induce autophagy in T98G glioma cells. A combination of sorafenib plus TMZ prevented autophagy induction by sorafenib and promoted apoptosis, mainly by downregulating the expression of Raf, Hsp27, and Hsp72, increasing ER stress [207]. Those authors suggested that the autophagy induced by sorafenib acts as a survival mechanism in T98G glioma cells [207]. Inhibiting autophagy with CQ or ATG5 siRNA improved the anti-neoplastic efficacy of sorafenib on U373 and LN229 glioma cells, inducing apoptosis, inhibiting migration and invasion in vitro and in vivo [277]. On the other hand, sorafenib was proved to induce apoptosis as a cell death mechanism. Yunhui et al. reported that sorafenib enhanced the anti-tumor effect of TTFields on U373 and U87 glioma cells in vitro and in vivo; it reduced cell proliferation, invasiveness, and angiogenesis, and enhanced apoptotic and autophagic cell death by ROS generation and a downregulation of markers related to epithelial to mesenchymal transition, such as vimentin and fibronectin [359]. A combination of sorafenib plus lapatinib (an ERBB1/2/4 inhibitor) act synergistically, killing glioblastoma cells by autophagy (ER stress) and apoptosis (via death receptors and mitochondria), inactivating ERK, AKT, mTOR, p70, and S6K, while decreasing the levels of Mcl-1 and Bcl-xL. Autophagy inhibition significantly decreased the rate of cell death induced by sorafenib and lapatinib [360]. Considering its activity as a kinase inhibitor, the use of sorafenib has been proposed in several clinical trials with glioblastoma patients. GBM cells usually show a high expression of VEGF and its receptors, an overactivation of PDGFR, mutations in Ras, and an overactivation of the MAPK pathway; this combination of markers indicates a poor prognosis. A combined administration of sorafenib plus temsirolimus (at a maximum dose of 800 mg/day and 25 mg/week, respectively) failed to show any improvement in the survival of patients with recurrent glioblastoma, while it had a worse toxicological profile. Lee et al. suggested that this poor result was due to the low dose of temsirolimus and the limited penetration of sorafenib into the central nervous system; thus, the effective dose of both drugs was not enough to inhibit the activation of multiple RTKs [361]. Schiff et al. also assayed the combined administration of sorafenib plus temsirolimus in patients with recurrent glioblastoma, but at higher doses: 200 mg twice a day for sorafenib and 20 mg/week for temsirolimus. Those authors found a significantly lower prevalence of adverse effects; however, the disease-free survival at 6 months (DFS-6) failed to improve; in fact, a slightly better result was observed in patients who did not received VEGF inhibitors [362]. Hottinger et al. assayed the security and efficacy of sorafenib in a phase-I study, establishing as safe a dose of 200 mg twice a day combined with the daily administration of 75 mg/m^2^ of TMZ. The main adverse effect was thrombocytopenia, which was more frequent with the combined treatment than with individual drugs. As an additional advantage, sorafenib did not alter the plasmatic concentration of TMZ [363]. In a phase-II study, Zustovich et al. co-administered 400 mg of sorafenib twice a day plus a daily dose of 40 mg/m^2^ of TMZ to a group of patients with relapsing glioblastoma until cancer progression resumed. A disease control rate of 55% was observed, with a mean OS of 7.4 months and DFS-6 of 26%; thus, sorafenib plus TMZ proved to be a safe combination with modest adverse effects in this group of patients [364]. Finally, combined treatments, such as sorafenib plus bevacizumab [365] and sorafenib plus tipifarnib [366] showed some advantage, but also high toxicity rates.

### 4.6. Targeting Downstream Intracellular Effector Molecules

#### 4.6.1. The RAS/RAF/MAPK Pathway

Farnesyltransferase inhibitors (FTIs) inhibit RAS-mediated signaling pathways (RAS/RAF/MAPK and RAS/PI3K/AKT) and Rho B by blocking the post-translational addition of a 15-carbon residue (farnesyl group) to the terminal cysteine residue of the CAAX tetrapeptide sequence in several proteins, particularly the p21Ras protein, coded by the *ras* oncogene. An overregulation of RAS has been widely described in GBM [367]. Lonafarnib and vemurafenib are FTIs approved to treat glioblastoma in the clinic (Figure 3).

#### 4.6.2. Lonafarnib

Lonafarnib and other FTIs induce cell death by autophagy in various solid tumor cells, and also inhibit caspase activity [368,369,370]. The pro-autophagic effect of lonafarnib (SCH66336) has been observed in several cell lines [371,372]. The mechanisms by which FTIs induce autophagy are still under study, but it has been proposed that its effect is due to an alteration in the Ras/PI3K/AKT/Rheb/mTOR pathway, along with ROS formation and damage to DNA. It is known that lonafarnib decreases mTOR phosphorylation and activation in a dose-dependent manner [368]. Lonafarnib inhibited cell proliferation in a panel of anaplastic astrocytoma cell lines and in mice with GBM xenografts [373]. Lonafarnib also inhibited cell proliferation and growth in a glioma cell line overexpressing EGFR, and induced arrest in the G_2_ phase of the cell cycle by reducing pMAPK levels [374]. A combination of lonafarnib plus cisplatin produced synergistic antiproliferative effects in T98G human glioblastoma cells [375]. On the other hand, lonafarnib boosted the cytotoxic effect of radiation and TMZ in orthotopic GBM tumors and neurospheres derived from human GBM tissue [376]. A phase-I and pharmacokinetic study of oral lonafarnib in pediatric patients with refractory brain tumors, including glioma and GBM, showed a good tolerability and safety, and especially a limited hematopoietic toxicity [377]. A combination of lonafarnib plus TMZ proved to be an attractive therapeutic strategy in malignant glioma and GBM [378,379]. Clinically, a lonafarnib-TMZ combined treatment showed a PFS-6 of 38% and a median PFS of 3.9 months. The median disease-specific survival was 13.7 months in 27% of recurrent glioblastoma patients [379]. Resistance to lonafarnib has been related to an activation of IGF-IR in vitro. Lonafarnib activated the IGF-IR/PI3K/Akt pathway, leading to increased mTOR-mediated protein synthesis in cell lines resistant to the apoptotic effect of the drug [380].

#### 4.6.3. Vemurafenib

Vemurafenib is a selective ERK inhibitor in BRAFV600-positive tumor cells [381]. Increased autophagy rates have been observed in the cells of tumors affecting the central nervous system with a BRAF V600E mutation; a combined treatment with vemurafenib plus autophagy inhibitors, such as CQ, significantly increased the chemosensitivity of these cells [382]. Levy et al. reported a synergic effect of vemurafenib plus CQ or a genetic autophagy inhibitor. The combined treatment significantly increased cytotoxicity and cell death by activation of caspase-3/7 in BRAF V600E-positive tumor cells. In BRAFV600E-positive tumor cells resistant to BRAF inhibitors, this combination induced chemotherapeutic sensitization and decreased cell growth rates by 50% [383]. Vemurafenib-resistant cells could be re-sensitized by downregulating BAG3, an antiapoptotic protein of the TSP family found in various tumor types, including glioblastoma [384]. A combined treatment with vemurafenib plus *bag3* siRNA significantly increased apoptosis rates in neoplastic cells resistant to BRAF inhibitors [385]. The triterpenoid saponin cumingianoside A (CUMA), a phytopharmaceutical extracted from the leaves of *Dysoxylum cumingianum*, has been proved to decrease the resistance to vemurafenib. A combination of vemurafenib plus CUMA reduced the rate of tumor growth, proliferation, and angiogenesis, inducing apoptosis by the mitochondrial route, ER-stress, and arrest of the cell cycle in the G_2_/M phase, both in vivo and in vitro [386]. BRAF V600-mutant cells are found in 1% of glioblastomas with epithelial histopathological features [387]. Vemurafenib showed a remarkable antineoplastic effect in high-grade, BRAFV600E-positive gliomas, with a PFD of more than 6 months and an overall clinical benefit rate of 38% [388]; it has also shown antiepileptic effects in BRAFV600E-positive brain tumors in vitro [389] and shown promising results on anaplastic pleomorphic xanthoastrocytoma, reducing nodular progression and edema at 3 months [390]. In brain stem ganglioglioma and anaplastic ganglioglioma, vemurafenib decreased tumor size and relieved the symptoms within months [391]. In another case report on anaplastic ganglioglioma, tumor resection was followed by vemurafenib administration; since BRAF inhibition induces a sustained activation of MAPK, which may contribute to drug resistance, Touat et al. co-administered vemurafenib plus cobimetinib, a MEK inhibitor. The patients receiving the combination remained asymptomatic for 16 months, showed a good tolerance to the treatment, and did not show recurrence [392]. Vemurafenib induced a tumor size reduction in BRAFV600E-positive metastatic glioma [393]. Similarly, the combined administration of vemurafenib plus everolimus to patients with BRAFV600E- and PIK3CA-positive extracranial metastatic anaplastic oligoastrocytoma delayed the progression of the disease for 6 months before chemoresistance was observed [222].

### 4.7. Inhibition of the PI3K/AKT/mTOR Pathway

The PI3K/AKT/mTOR pathway plays a key role in the proliferation, survival, and migration/invasion of glioblastoma cells. Drugs known to inhibit mTOR, such as temsirolimus, sirolimus, and everolimus, are currently being assayed in clinical trials (Table 2).

#### 4.7.1. Temsirolimus

Temsirolimus (CCI-779) is a lipid-soluble rapamycin analogue that binds the FK-binding protein 12 (FKBP-12), inhibiting mTORC1 [394,395]. Temsirolimus showed an antiproliferative effect on mouse low-grade glioma, GBM [396], and U87-MG glioma cells [397]. Temsirolimus modulates the NF-κB and PKC-α signaling pathways [398], inhibits the expression of mesenchymal markers, and repress stem-like cell properties in GBM cells [399]. Due to the limited activity of temsirolimus in glioma and GBM as a single agent, several trials have been conducted using multimodal therapies. Combined with perifosine (an AKT inhibitor), temsirolimus showed antiproliferative and antiapoptotic effects disregarding the PTEN status, correlating with a decrease in the PI3K/AKT/mTOR activity [400,401]. In vitro, a combination of the mitochondrial inhibitor PENAO plus temsirolimus induced mitochondrial dysfunction, showed an enhanced cytotoxic activity, inducing apoptosis, and decreased the clonogenic ability of diffuse intrinsic pontine glioma cells [402]. Diffuse intrinsic pontine glioma cells treated with palbociclib plus temsirolimus synergistically decreased cell proliferation, likely due to the loss of metabolic compensatory pathways [403]. A combination of radiotherapy plus temsirolimus improved survival rates in a syngeneic mouse glioma model through additive cytostatic effects; it also showed anti-proliferative and anti-invasive effects [404]. Temsirolimus was well tolerated in trials on GBM patients, but its antitumoral efficacy has been modest [394,405]. In a phase-I, dose-escalation trial, the co-administration of temsirolimus plus radiotherapy and TMZ to patients with newly diagnosed GBM was associated with several adverse events, including increased infectious risks [406]. In a phase-II study on GBM patients, temsirolimus showed no antineoplastic effects, either as a monotherapy or combined with erlotinib [304]. Temsirolimus was not superior to TMZ in patients with an unmethylated *MGMT* promoter in glioblastoma [407]. Phase-II studies with recurrent glioblastoma have reported no efficacy of temsirolimus, neither alone nor combined with TMZ, sorafenib, bevacizumab, nor erlotinib [362,408]. Recurrent GBM patients showed no radiological response after a temsirolimus-bevacizumab combined treatment (median PFS was 8 weeks, and OS was 15 weeks) [409]. Temsirolimus is well tolerated in pediatric patients [401,410]. Patients with refractory glioblastoma receiving temsirolimus plus bevacizumab showed a PFS of 4 months [411]. The combined administration of temsirolimus plus sorafenib did not improve the survival of recurrent GBM patients either [361]. Resistance to mTORC1 inhibitors, such as temsirolimus involves an upregulation of glutamine metabolism, PTEN expression, alteration of the p53 response to nucleolar stress, a negative feedback activation of the PI3K/AKT signaling, increased rates of clonal and non-clonal chromosome aberrations, and changes in the copy number alterations pattern [412,413,414]. The NCT Neuro Master Match (N2M2) trial is an ongoing open-label, multicenter, phase-I/IIa umbrella trial for glioblastoma patients with or without *MGMT* promoter hypermethylation. Based on molecular findings, the patients were allocated in different subtrials, receiving temsirolimus, alectinib, palbociclib, idasanutlin, vismodegib, asinercept, atezolizumab, or TMZ. The results of the trial have not been published yet, but the frequency of adverse effects and the efficacy of each subtrial will be reported. The study will provide the basis to develop predictive biomarkers and select the best-suited treatment for each patient [415,416].

#### 4.7.2. Everolimus

Everolimus, a rapamycin derivative (also called RAD001) is an orally administered mTOR inhibitor with activity against various tumor types, included glioblastoma [396,417]. Everolimus decreases the expression and activity of lactate dehydrogenase and choline kinase α, reducing the levels of lactate and phosphocholine by transcriptional inhibition of HIFα in glioma models, both in vitro and in vivo [418]. Since an overexpression of the protein MCL-1 has been related with drug resistance [419], a depletion of MCL-1 by everolimus can suppress the homologous recombination repair in GBM, increasing DNA damage [420]. The scavenger receptor class B member 1 (SCARB1) is highly expressed in LN229 cells compared to U87 cells; therefore, the cytotoxic effect of everolimus is significantly higher (185 times) in LN229 cells when transported by high-density lipoprotein (HDL) nanoparticles, while free everolimus is only 3.2 times more effective against U87 cells [421]. The combined administration of everolimus plus palbociclib, a specific CDK4/6 inhibitor that prevents DNA synthesis by arresting the cell cycle in the G_1_ phase, has shown a synergistic effect on GBM (in vitro and in vivo) by inducing apoptosis. In addition, everolimus improved the penetration and bioavailability of palbociclib in the CNS [422]. Everolimus and LBT613 (a RAS inhibitor) exhibited a synergistic effect to inhibit cell proliferation, migration, and invasion in glioma cell lines (U87-MG and U373MG) [423]. The combined administration of everolimus plus AEE788 (an EGFR and VEGFR inhibitor) was more efficient in inducing apoptosis and cell cycle arrest than either drug alone, both in vitro and in vivo [424]. The Delta-24-RGD adenovirus exerted a synergistic effect on U87-MG human glioma cells when co-administered with everolimus, improving its antitumor activity. In experiments in vitro and in vivo, a combination of both agents induced autophagy by promoting the expression of Atg5, a key protein to transform LC3-I into LC3-III; this treatment increased the mean survival of mice with glioma by 80% [425]. Everolimus significantly potentiated the effect of TMZ to induce cell death by autophagy in the U87 glioma cell line [410]. The use of mTORC1/mTORC2 dual inhibitors (everolimus and AZD2014) plus olaparib (an inhibitor of poly(ADP-ribose) polymerase) was recently reported to induce necroptosis and reduce chemoresistance in GBM and colorectal carcinoma cells (expressing a mutated PTEN) [420]. In a phase-I study, the combined administration of everolimus plus radiotherapy/TMZ or TMZ as an adjuvant was well tolerated by GBM patients [426]. However, the results of a phase-II clinical trial failed to show improvement in patient survival, despite the increased cytotoxicity of the combined treatment [427,428]. The administration of everolimus plus standard chemotherapy in a pediatric case of giant-cell GBM expressing TP53 and R611W mutations in TSC2 resulted in a complete remission and survival at 33 months, with an excellent quality of life [429]. GBM patients showed a median PFS of 11.3 months and a median OS of 13.9 months after receiving the standard therapy (radiotherapy/TMZ) supplemented with everolimus and bevacizumab [430]. In contrast, a combination of gefitinib plus everolimus failed to show therapeutic efficacy with respect to the historical control, even though a radiological response was observed in 33% of recurrent glioblastoma patients [431].

### 4.8. Novel and Promising Strategies in Pre-Clinical Stages

Due to the lack of effective surgical and medical treatments for glioblastoma, novel promising alternatives targeting autophagy are being developed. However, before these new therapies can be used in the clinical practice, their effects must be re-evaluated in controlled clinical trials to corroborate their real effectivity. These strategies include miRNAs, BH3 mimetics, and cannabinoids, as well as histone deacetylase and proteasome inhibitors.

#### 4.8.1. MicroRNAs

MicroRNAs (miR) are non-coding, single-stranded, 17–25 nucleotide-long RNA fragments that regulate biological processes such as proliferation, invasion, metabolism, apoptosis, and autophagy through post transcriptional gene silencing [432]. It has been reported that miR are dysregulated in various malignancies and that they can act either as tumor suppressors or oncogenes [433]. Jiang et al. showed that the long non-coding RNA CASC2 is underexpressed in glioma cells, thus favoring an increase in miR-193a-5p and a decrease in mTOR activity, which induces protective autophagy, leading to TMZ resistance. Inhibiting autophagy by miR-193a-5p inhibition increases the efficacy of TMZ by downregulating LC3-II and Beclin-1 [218]. miR-128 was reported to directly block the mTOR, RICTOR, IGF1, and PIK3R1 pathways, activating apoptotic and autophagic cell death in U87 glioma cells [217]. miR155-3p also induced the IL-6-STAT3 signaling pathway, enhancing hypoxia-induced autophagy through the CREB3 regulatory factor (CREBRF)-cAMP responsive element binding protein 3 (CREB3) and the ATG5 pathway in U251 and T98G glioma cells. Inhibiting IL6 by tocilizumab induced apoptosis in glioma cells [434]. Huang et al. reported that the expression of miR-93 regulated autophagy in glioma stem-like cell by simultaneous inhibition of autophagy regulators, such as Beclin-1, ATG4B, ATG5, and p62, decreasing cell growth and glioma sphere self-renewal in vitro, and tumor cell growth in vivo [216]. Suppressing autophagy through the ectopic expression of miR- 93 or by administering CQ sensitized glioma stem-like cell to TMZ and IR therapy. IR, TMZ, and rapamycin have been reported to decrease the levels of miR-93 [216]. An overexpression of miR-224-3p in U251 and U87 glioma cells increased hypoxia-induced apoptosis by inhibiting the expression of ATG5 and FIP200, while inhibiting tumorigenesis in GBM cells in vivo [125]. miR-517c suppressed autophagy and decreased tumor invasion in glioma cells [435]. A better understanding of the underlying molecular mediators (i.e., miRNAs) and their functions in autophagy pathways could help us to develop novel treatment approaches for patients with brain tumors [436].

#### 4.8.2. BH3 Mimetics

The anti-apoptotic proteins Bcl-2, Bcl-xL, and Mcl-1 participate in the inhibition of both apoptosis and autophagy and are involved in the resistance to therapy in various malignancies, including malignant gliomas. BH3 mimetics, such as ABT-737 and gossypol, being the later the most active antineoplastic agent in the group, were developed to specifically target Bcl-2 members. (−)-Gossypol, an inhibitor of Bcl-2, Bcl-xL, Bcl-w, and Mcl-1, significantly promoted cell death by apoptosis in several glioma cell lines [437]. (−)-Gossypol also decreased growth rates in TMZ-resistant glioma cells [438]. Voss et al. reported that gossypol significantly induced autophagic cell death independent of caspase activity in glioma cell lines [439]. A knockdown of Beclin-1 and Atg5 reduced the rates of cell death, and this effect was reverted by an mTOR knockdown. These results indicate that autophagy contributes to this type of cell death [439]. In addition, a gossypol/TMZ combination inhibited multiple cancer hallmarks, such as tumor proliferation, angiogenesis and invasion by promoting apoptosis in several GBM models [440]. Recently, (−)-gossypol was reported to induce cell death by mitophagy, increasing the levels of BNIP3 and BNIP3L in glioma cells [441]. These results suggest that pan-Bcl-2 inhibitors, either alone or combined with other therapeutic agents, are promising drugs in apoptosis-resistant tumors, such as glioma.

#### 4.8.3. Cannabinoids

Some derivatives of Cannabis sativa have shown an antineoplastic effect on glioma by activating autophagic mechanisms. Δ9-tetrahydrocannabinol (THC) induced ER stress through dihydroceramide accumulation and the phosphorylation of the eukaryotic translation initiation factor 2α (eIF2α) [442,443]. This leads to increased levels of the proteins p8 and TRB3, which inhibit mTORC1 and activate autophagy [443,444]. Thus, cell death by autophagy would be beneficial for glioma patients. Some glioma cells seem to be resistant to cannabinoid treatment; this effect could be linked to the growth factor midkine (Mdk), which inhibits TRB3 and phosphorylates ALK, thus inhibiting the autophagic cell death induced by THC [445,446]. The effect of TMZ plus THC was assayed in glioma xenografts, observing a decrease in tumor size that correlated with autophagy activation [235,447]. The cannabinoid β-caryophyllene (BCP) was recently assayed in glioma models. BCP binds the cannabinoid receptor 2 (CB2), avoiding the psychotropic effects produced by CB1 stimulation. Irrera et al. proposed that BCP decreases the expression of Beclin-1, LC3, and p62/SQSTM1, suggesting that autophagy was inhibited, and that cell programming was switched to apoptosis; the result was a decrease in cell viability [448]. Another study reported that cannabidiol (CBD) induces cell differentiation and cell death by autophagy in a model of glioblastoma stem-like cells (GSCs). Interestingly, CBD activated the transient receptor potential vanilloid-2 (TRPV2) channel and induced autophagy. TRPV2 is a target for the acute myeloid leukemia 1 a splice variant (Aml-1a), which is upregulated by CBD, leading to cell differentiation proliferation inhibition by autophagy [449].

#### 4.8.4. Histone Deacetylase Inhibitors

Histone deacetylase (HDAC) inhibitors promote histone acetylation, leading to changes in chromatin conformation and gene expression. Gonçalves et al. showed that CQ potentiated the anti-neoplastic effects of TMZ and SAHA in vitro [214]. CQ improved the efficacy of SAHA and TMZ to induce apoptosis due to ROS generation and the accumulation of damaged mitochondria [450]. On the other hand, SAHA acts as inhibitor of mTOR, increasing autophagy [451]. Chiao et al. reported that blocking ATG5, LC3, or Beclin-1 by shRNA and PI3K-III inhibitors, such as wortmannin and 3-MA, promoted apoptotic death cell induced by SAHA in glioma stem cells [452]. Moreover, ATG7 depletion potentiated the apoptosis induced by SAHA in T98G glioma cells [453]. In addition to SAHA, the HDAC6 inhibitor J22352 inhibits the fusion of autophagosomes with lysosomes, resulting in metabolic stress followed by autophagic death of cancer cells [454].

#### 4.8.5. Proteasome Inhibitors

The ubiquitin proteasome system (UPS) is the main pathway for the degradation of short-lived proteins in cells. Due to their ability to regulate cell death in neoplastic cells, proteasome inhibitors were assayed for their potential as cancer therapeutic agents, either in monotherapy or combined with other treatments. The proteasome inhibitor MG-132 inhibited cell growth, induced cell cycle arrest at the G_2_/M phase and led to death cell by autophagy in glioma cells [455]. Meanwhile, autophagy inhibition by 3-MA increased the rate of death cell induced by MG-123 [455]. Zhang et al. reported that the inhibition of autophagy by 3-MA or Atg7 siRNA led to significantly higher rates of cell death by treatment with bortezomib in U87 and U251 glioma cells [456]. Fang et al. showed that bortezomib induced autophagy by downregulating TRAF6 [457]. The activity of TRAF6 ubiquitin ligase was reported to induce the ubiquitination of ULK1 in Lys63, blocking autophagy [458].

## 5. Conclusions

Glioblastoma multiforme (GBM) is one of the most difficult neoplasms to treat due to its heterogeneity, tumor microenvironment, and invasiveness, which allows neoplastic cells to infiltrate the brain parenchyma, preventing a total resection of the tumor. In addition, GBM contains glioma stem cells, which show a great proliferation and self-renewal capacity, critically contributing to the tumor onset and therapeutic resistance. It has been suggested that pharmacologic resistance in GBM is due to a high concentration of antiapoptotic proteins (Bcl-2, Bcl-xL and Mcl-1) and a low expression of proapoptotic proteins (BAX, Bak, pBad, pBim, and Survivin), favoring the switching-on of oncogenes and genetic instability that promote tumor survival and resistance to radiotherapy, chemotherapy, and immunotherapy. Cell death by autophagy has emerged as a promising strategy to remove neoplastic cells. The modulation of the autophagic process as a death mechanism in neoplastic cells has led to the use of autophagy inhibitors and inducers to block cancer progression. Clinically, both autophagy inducers and inhibitors have been used, either as a monotherapy or combined with tyrosine kinase inhibitors such as EGFR, PDGR, and VEGFR, as well as EGFR/PI3K/AKT inhibitors. These treatments only have managed to extend the survival of GBM patients, but after an initial response, the antitumor effect is transient, and most tumors eventually progress with a high migratory and invasive capacity, followed by recurrence and resistance to various therapies. The lack of effectivity or cure, and the adverse effects of these drugs are mainly due to a dual effect of autophagy, which may be both a tumor-suppressing mechanism and a tumorigenic inducer, since an increase in autophagy may be beneficial by preventing tumor formation and progression, while autophagy inhibition could be advantageous to promote a complete regression. Thus, it is not easy to discern to what extent autophagy results in a positive or negative balance for cell survival, and therefore it is crucial to interfere with autophagy in a specific manner, adapted to the molecular, cellular, and epigenetic context, to the heterogeneity of tumor cells and microenvironment, and to the location of high-grade gliomas. Future translational studies on the interrelation and regulation of autophagy in tumor cells, glioma stem cells, and tumor microenvironment in a genic, epigenetic, and metabolic level will allow us to manipulate the complex processes of autophagy to devise new therapies (more selective inhibitors of autophagic processes) for high-grade gliomas.

## Figures and Tables

**Figure 1 pharmaceuticals-13-00156-f001:**
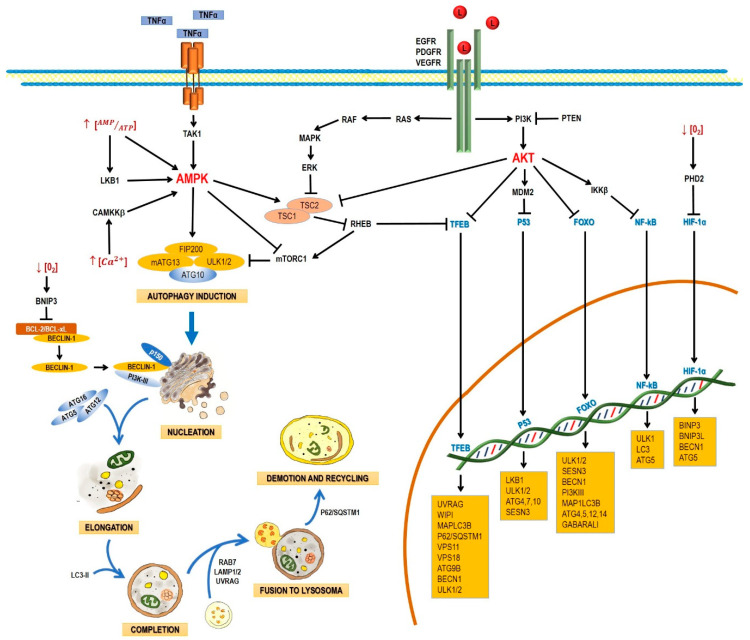
Schematic process of the autophagic pathway. Autophagy consists of the following stages: induction, phagophore nucleation, elongation, and completion; fusion to lysosome, degradation, and recycling. Receptor tyrosine kinases (EGF (epidermal growth factor receptor), PDGF (platelet-derived growth factor receptor), and VEGF (vascular endothelial growth factor receptor)) stimulate the RAS/RAS/ERK and PI3K/AKT signaling pathways, which traduce signals to activate transcription factors, such as transcription factor EB (TFEB), FOXO, P53, and NF-кB, which regulate the expression of genes that are key for autophagy induction. Then, the activation of AKT inhibits autophagic death by activating the mTORC1 complex, which contains mTOR. During nutrient deprivation, AMPK stimulates autophagy by phosphorylation of ULK1 and suppression of mTORC1, which inhibits autophagy by inactivating ULK1. Autophagy can also be induced by hypoxic conditions through HIF activation. Autophagosome biogenesis starts with the formation of an initiation membrane, derived from the endoplasmic reticulum (ER) or other cellular membrane sources. Vesicle nucleation requires the activity of the PI3K-III/Beclin-1 complex. The completion and expansion of the autophagosome requires the Atg5/Atg12/Atg16 and LC3-PE proteins. Mature autophagosome fuses with a lysosome to form the autophagosome, which requires the action of proteins, such as Rab7, UVRAG, and Lamp 1, 2; finally, p62/SQSTM1 is required for the degradation and recycling of the cellular components. Black arrows (↓) indicate activation, and black truncated arrows (⊥) indicate inhibition. Blue arrows (↓) indicate the process of the autophagic pathway.

**Figure 2 pharmaceuticals-13-00156-f002:**
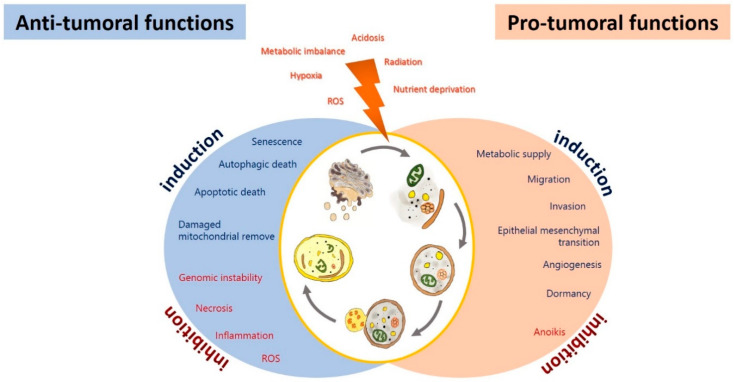
Anti-tumoral and pro-tumoral functions of autophagy. As an anti-tumoral process, autophagy prevents the accumulation of p62 aggregates, damaged proteins, and mitochondria, which may lead to ROS generation and the activation of oncogenic signaling pathways to stimulate necrosis, inflammation, and genomic instability, which lead to malignant transformation, tumor cell proliferation, migration, and invasion. Excessive autophagy can lead to type-II programmed cell death, which can induce apoptosis and activate senescence. On the other hand, autophagy sustains tumor growth and survival under adverse conditions (hypoxia and metabolic, osmotic, and oxidative stress). Uncontrolled proliferation leads to degrading misfolded or unnecessary proteins and organelles to mobilize amino acids, lipids, and carbohydrates, promoting cell survival. It also increases oncogenic signals that favor metabolism (glycolytic functions), angiogenesis, migration, and tumor invasion; additionally, it inhibits anoikis and dormancy.

**Figure 3 pharmaceuticals-13-00156-f003:**
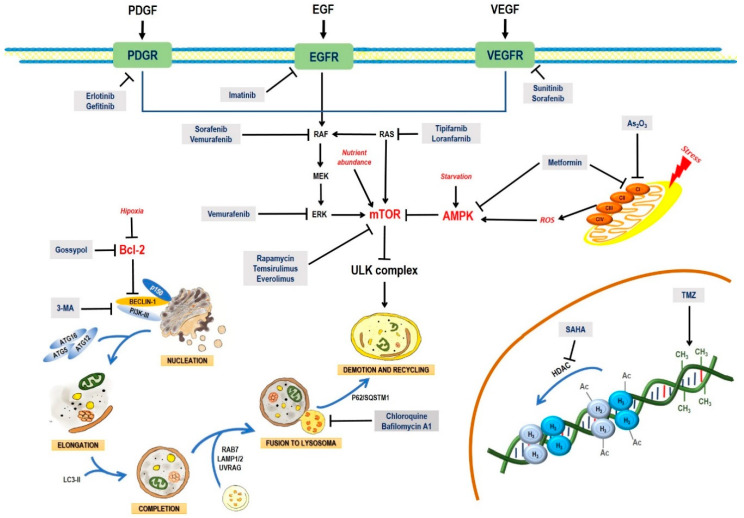
Molecular targets of antineoplastic agents in autophagy signaling pathways. Erlotinib and gefitinib (EGFR inhibitors), imatinib (PDGFR inhibitor), sunitinib and sorafenib (VEGF inhibitors), Tipifarnib and lonafarnib (farnesyl transferase inhibitors), sorafenib and vemurafenib (RAF and ERK inhibitor). Metformin inhibits AMPK and the Complex I (CI) of the mitochondrial electron transport chain; CI is also inhibited by AS_2_O_3_, rapamycin, temsirolimus, and everolimus (mTOR inhibitors), gossypol (Bcl-2 inhibitor), 3-MA (PI3K-III), chloroquine, and bafilomycin A1 (autophagy inhibitors), SAHA (HDAC inhibitor), and TMZ (alkylating agent). Black truncated arrows (⊥) indicate inhibition.

**Table 1 pharmaceuticals-13-00156-t001:** Effect of temozolomide in glioma.

Treatment/Pharmacologic Category		Mechanism of Action	Clinical and Pre-Clinical Trials
**Temozolomide** **(TMZ)** (Second-generation alkylating agent analogue to mitozolomide)		Active metabolite 3-methyl-(triazen-1-yl) imidazole-4-carboxamide; erroneous matching of O6-methylguanine with thymidine during DNA replication; leads to a halting in the G2/M phase of the cell cycle and subsequently to cell death [202].	Combined with radiation it shows increase in median OS from 12.1 months to 14.6 months [201].
**TMZ in combination with other treatments**	**Sorafenib** (Multikinase inhibitor)	Induces apoptosis independent to caspase way in MOGGCCM cells and caspase-dependent apoptosis in T98G cells; the inhibition of Ras-Raf-MEK- ERK pathway increases sensitivity to apoptosis in both cells lines [207].	Sorafenib in combination with TMZ acts in synergy and triggers cell death by apoptosis in GBM (T98G) and anaplastic astrocytoma (MOGGCCM) cells [207].
	**Momelotinib** (Inhibit to Janus kinase (JAK)-1/2)	Inactivation of JAK2/STAT3 pathway improves apoptosis and autophagy; reduction the MGMT expression; downregulates Ki-67, PCNA, p-STAT3, and p-JAK2; increases the expression of cleaved caspase-3 and caspase-9, LC3-II and Beclin 1; arrests the cell cycle in the G_2_ phase [208].	A combination of both drugs increases cell death by apoptosis and autophagy in U251 cells and sensitizes GBM cells to TMZ [208].
	**Roscovitine** (Cyclin-dependent kinase Cdk inhibitor)	Inhibition of Cdk5 increase autophagy and apoptosis dependent of caspase-3; decreases the levels of Ki67, GFAP, VEGF, and CD31; reduces tumor progression by decreasing reactive astrocytes surrounding the glioma [209].	The combination prevents cell growth of glioma cells, angiogenesis, and proliferation; it sensitizes to TMZ resistant U87, U373, LN18, and C6 cells [209].
	**MSNP-TMZ-PDA-NGR plus 3-methyladenine (3-MA)** (Controlled release of TMZ in cells with overexpression of CD13; 3-MA is an inhibitor of PI3K that avoid autophagy)	Strong increase in caspase 3-dependent apoptosis [210,211].	Causes apoptosis specifically in neoplastic cells (C6) and neovascular endothelial cells [210,211].
	**BIX01294** (Histone methyltransferase G9a inhibitor)	Reduces methylation of H3K9 and H3K27, favoring the activation of caspases to induce apoptosis; increases LC3-II levels in LN18 cells [212].	The treatment sensitizes TMZ in both mutated and wild type PTEN in LN18 and U251 glioma cell lines and glioma stem cells [212].
	**Tubacin** (Histone deacetylase 6 (HDAC6) inhibitor)	Increases the accumulation of LC3B-I and LC3B-II, p62 and autophagosome formation; blocks the fusion of autophagosome and lysosome [213].	Inhibits glioma cell growth; blocks autophagy; increases the cytotoxicity of TMZ in U251 and LN229 line cells [213].
	**Suberoylanilide hydroxamic acid** **(SAHA)** (Histone deacetylase inhibitor)	Arrests the cell cycle in G_2_/M phase; induces autophagy and apoptosis; the cytotoxicity of SAHA/TMZ is improved by inhibition of autophagy with chloroquine [214].	Potentiates the effect and cytotoxicity of TMZ in both C6 and U251-MG cells; SAHA/TMZ/CQ intensifies apoptosis and TMZ sensitivity [214].
	**miR-93** **overexpression** (Downregulates the expression of proteins involved in autophagy)	Autophagy inhibition by suppressing the expression of BECN1, ATG5, ATG4B, and SQSTM1; the synergic effect of miR-93 and the ATG4B antagonist NSC185058 blocks autophagy and increase TMZ-induced apoptosis; the double inhibition of autophagy by miR-93 plus CQ exacerbate the TMZ cytotoxicity and then the cell apoptosis [216].	The inhibition of autophagy sensitized GSC (MES GSC 83 cells) to TMZ and provoke cell apoptosis.
	**miR-128** (Regulates cell death and survival genes)	TMZ induces the expression of miR-128 through the signaling pathway JNK2/c-Jun; overexpression of miR-128 promotes apoptosis by enhancing caspase-3/9 activation, ROS generation, MMP loss, downstream HIF-1, and non-protective autophagy through inhibition of the mTOR pathway [217].	Overexpression of miR-128 enhanced TMZ apoptosis and non-protective autophagy in U87-MG cells [217].
	**Long non-coding** **RNAS** **cancer susceptibility candidate** **2 (CASC2)** (Tumor suppressor)	CASC2 modulates mTOR-dependent autophagy induced by TMZ; inhibition of autophagy by CASC2 sensitized glioma cells to TMZ cytotoxicity [218].	CASC regulates the resistance induced by TMZ and decreases tumor cell survival, migration, and invasion in U257-TR and U87-TR (TMZ-resistant) [218].
	**miR-519a** **overexpression** (Tumor suppressor by regulation of the STAT3 pathway in glioma)	miR-519a inhibits STAT3/Bcl2 signaling and acts synergistically with TMZ to enhanced apoptosis mediated by autophagy [208].	Increases sensitivity to TMZ and promotes autophagy and apoptotic cell death in GBM cells (U87-MG) [208].
	**miR-224-3p** **overexpression** (Regulates hypoxia-induced autophagy)	Downregulates hypoxia-induced autophagy by ATG5 inhibition and sensitizes glioma cell to TMZ [220].	Blocks hypoxia-dependent autophagy, increases TMZ chemosensitivity and prevents cell mobility in LN229 under hypoxia [220].
	**Endothelial-monocyte- activating polypeptide****-II (EMAPII)** (Secretory polypeptide protein, with angiogenesis and pro-apoptosis effect)	Upregulates the expression of the Bcl-2/adenovirus E1B 19 kDa protein-interacting protein 3 (BNIP3) by downregulating the expression of miR-24-3p; stimulates mitophagy mediated by BNIP3 [218].	EMAP-II triggers the cytotoxic effect of TMZ, induces mitophagy, inhibits cell viability, migration, and invasion in U87-GSCs and U251-GSCs [218].
	**Cation transport regulator-like protein** **1 (CHAC1)** (Pro-apoptotic proprieties)	Inhibits Notch3, increases intracellular Ca^2+^ and ROS generation, and loss of MMP [221].	Increases sensitivity to TMZ, decreases cell proliferation and migration in U87-MG cells [221].
	**GDC-0941** (Specific pan-PI3K inhibitor)	Reduces p-AKT and enhances the expression of GSK3β and p53. Increases apoptosis, autophagy, and TMZ cytotoxicity [222].	Promotes apoptosis, autophagy, and TMZ cytotoxicity in T98G, SHG44, and A172 glioma cells [222].
	**Ionophores nigericin and salinomycin** (Intracellular Ca^2+^ and K^+^ mobilizing)	Promotes ROS generation to cause cell death; suppresses glioma spheroid formation [223].	Combination treatment with an inhibitor of autophagy induces cell death in KGS01-KGS03 and TGS01-TGS04 cells lines [223].
	**Guanosine** (Neuroprotective agent with the propriety to mediate the glutamatergic system)	Increases autophagy by acidic vesicular organelles; reduces the mitochondrial membrane potential and induces apoptosis; modulates the activity of adenosine receptors [223].	Combination treatment alters mitochondrial function and increases apoptosis; decreases cell migration, proliferation and growth in A172 glioma cells [223].
	**Atorvastatin** (Reduces the biosynthesis of cholesterol, regulates abnormal lipid storage, has anti-inflammatory and neuroprotective effects)	Cytotoxic effect dependent of glutamatergic receptors activation; induces apoptosis and autophagy [223].	Limits cell proliferation and migration; it reduces cell viability in A172 glioma cells [223].
	**Thioridazine** (Antipsychotic drug with antineoplastic effects)	Triggers the activity of AMPK and LC3, and P62 expression; downregulates Wnt/β-catenin signaling; increases Bax activity and decreases Bcl-xL expression; induces autophagy-dependent apoptosis by a p62-dependent mechanism and caspase 8 involvement, and apoptosis through a P53-independent pathway [224].	Decreases cell viability by inducing apoptosis and autophagy in U87-MG and GBM8401 cells [224].
	**Tuibolone and/or medroxyprogesterone acetate (MPA)** (Hormone replacement therapy, analog of progesterone)	Suppresses DNA synthesis; produces autophagosome and autophagic vacuoles; alters mitochondrial activity and lysosomes [224].	Combination treatment acts against clonogenic growth of C6 glioma spheroids by inducing mitophagy and autophagy [224].
	**Β-****asarone** **(CIS-2,4,5-****trimethoxy****-1-****allyl phenyl)** (Polyphenols extracted from *Acorus tatarinowii* Schott and *Guatteria gaumeri* Greenman)	Arrests the cell cycle in the G_0_/G_1_ phases; enhances P53, LC3-II/I, Beclin-1, AMPK, and pAMPK levels; prevents P62, Bcl-2 and mTOR expression; induces autophagy via P53/Bcl-2/Beclin-1 and P53/AMPK/mTOR [226].	Inhibits cell growth and promotes autophagy in U251 glioma cells [226].
	**Honokiol** **(2****-(****4-****hydroxy****-3-****prop****-2-****enyl-phenyl)-****4-****prop****-2-****enyl-phenol)** (Polyphenol extracted from *Magnolia officinalis* with anti-neoplastic effect)	Enhances LC3-II expression; induces ROS production and activates caspase-3; increases the levels of acidic vesicular organelles; arrests cell cycle in G_1_ phase and produces DNA fragmentation; promotes autophagy by activation of the p53/PI3K/Akt/mTOR signaling pathway; induces apoptosis dependent of p53/cyclinD1/CDK6/CDK4/E2F1, and autophagy followed by apoptosis; sensitizes TMZ-resistant cells [226,227,228].	Decreases cell proliferation by inducing apoptosis and autophagy in glioma U251 cells [227,228].
	**Carnosic acid** (Polyphenolic from *Rosmarinus officinalis* or *Salvia officinalis* with anticancer properties)	Arrests the cell cycle in G_0_/G_1_ phase; decreases the mitochondrial membrane potential; improves poly (ADP-ribose) polymerase segmentation; increases the segmentation of caspase-3 and decreases the levels of Cyclin B1; increases the levels of LC3-II and reduces p62 expression; prevents the blockage of autophagy by inhibition of the PI3K/AKT signal pathway [239].	Combination synergistically increases the TMZ cytotoxic effect and prevents cell migration by inducing apoptosis and autophagy in U251 and LN229 [230].
	**Curcumin** (Polyphenol isolated from rhizome of *Curcuma longa* with anti-angiogenic and anti-proliferative effects)	Causes alteration in the EGFR/PI3K/PTEN/RAS/STAT-3 pathways; prevents gliomagenesis by decreasing connexin 43 expression [231,232].	Sensitizes TMZ-resistant cells and induces apoptosis [231,232,233].
	**Euphol** **(A** **tetracyclic triterpene alcohol)** (Phytopharmaceutical with analgesic, anti-inflammatory and antiviral properties, isolated from *Euphorbia tirucalli*)	Improves the TMZ cytotoxic effect with highest sensitivity in pediatric gliomas; enhances the cytotoxicity by autophagy inhibition with Bafilomycin A1; induces autophagy-associated protein LC3-II and acidic vesicular organelles; promotes cell cycle arrest at the G_0_/G_1_ phase; antiapoptotic effect [234].	Arrests cell proliferation and migration by inducing autophagy; it shows antitumoral and antiangiogenic effects [234].
	**Δ9-****tetrahydrocannabinol** **(THC)** **and cannabinol** **(CBD)** (Cannabinoids extracted from *Cannabis sativa*)	Decreases Ki67 expression; increases LC3 levels; causes DNA fragmentation; induces autophagy and apoptosis [235].	Shows a significative effect on cell proliferation; induces cell death dependent of autophagy and apoptosis in U87-MG glioma cells [235].

**Table 2 pharmaceuticals-13-00156-t002:** Treatment options of autophagic regulators in glioma.

Treatment/Pharmacologic Category	Mechanism of Action In Vitro	Clinical Trials
**Arsenic trioxide** 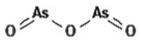 Inorganic compound; molecular formula: As_2_O_3_	ROS production: inhibiting the complex I of the respiratory chain and antioxidant enzymes (glutathione peroxidase and thioredoxin reductase), as well as activating NADPH oxidase [245]. Inhibits PI3K/AKT and NF-κB pathways [254,255,256]. Induces apoptosis through of the generation of ROS, which promote the downregulation of Bcl-2 and upregulation of Bax [246,247,248]. Induces apoptosis through of the p53 stability, expression of Fas, FasL and Bax and activation of caspase 3 and 9 [249]. Blocks the Notch signaling pathway in stem cell-like cancer cells [250], and plus AT101 (inhibitor of Bcl-2, Bcl-xL and Mcl-1 pro-apoptotic proteins) [251]. Increases caspase 3, inhibits PTCH1b, N-Myc, and GLI2 (transcriptional targets of Hg pathway), and inhibits HES5 and HEY1 (transcriptional targets of Notch pathway) [252]. Induction of apoptosis by inhibition of hERG channel expression, probably by upregulation of SRF-dependent miR- 133b [253]. Inhibits the activity of cathepsin B, MMP2, MMP9, and telomerase; induces the activity of caspase 3, and apoptosis. [256]. Induces mitochondrial damage and autophagy by overexpression of the mitochondrial cell death protein BNIP3 [260]. Induces G2/M cell cycle arrest and autophagic cell death [261]. Plus Bafilomycin A1 increases the rate of cell death by apoptosis [256]. Plus Chloroquine induces both autophagy and apoptosis by the action of Beclin-1 [263].	Plus TMZ and radiotherapy: median age 52 y (range: 25–80); overall survival (OS), 37 months; progression-free survival (PFS), 13 months (range: 4–57 months) [240]. Plus radiotherapy: median age 9 y (range: 3–19); OS, 13 months (range: 11–33) [241].
**Chloroquine (CQ)** 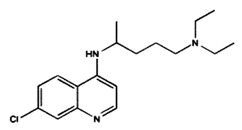 Synthetic antimalarial agent; molecular formula: C_18_H_26_ClN_3_ [4-*N*-(7-chloroquinolin-4-yl)-1-*N*,1-*N*-diethylpentane-1,4-diamine]	Prevents the autophagosome-lysosome fusion, inhibiting the final stage of autophagy and inducing cell death by apoptosis [266]. Synergizes the cytotoxic effect of TMZ by halting PI3K-(III)-Beclin-1 and GRP78, as well as the conversion of LC3- I into LC3-II; increases the expression of proapoptotic proteins, such as C-EBP, CHOP/GADD, and PARP [271]. Induces ROS generation and inhibits mitochondrial autophagy [272]. Increased the apoptosis induced by TMZ via p53. the antineoplastic response to the CQ-TMZ combination depends on the status of p53 [266] Plus radiotherapy: promotes cell cycle arrest in the G_0_/G_1_ phase, increases caspase 3 expression, and decreases Bcl-2 expression and autophagosome [266]. Plus sorafenib: Increases anti-proliferative and anti-invasive effects [275]. Enhances the antineoplastic effect of vorinostat (inhibitor of histone deacetylases) [214].	Increased survival time to 25 ± 3.4 months with respect to the control group (11.4 ± 1.3 months) [270]. Plus carmustine and radiotherapy: improved median survival from 5 to 10 months, and 3 to 15 months in EGFRvIII-negative and EGFRvIII-positive GBM, respectively [267] Plus TMZ and radiotherapy: survival time was 35 ± 5 months [268]. Mean survival post-surgery is 24 months [269].
**Metformin** 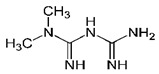 Biguanide type, obtained from the extract of *Galega officinalis* [276]; molecular formula: C_4_H_11_N_5_ [3-(diaminomethylidene)-1,1-dimethylguanidine]	Decreases glucose level in blood [276]. Reduces cell proliferation by the downregulation of the AKT pathway; arrests cell cycle associated to JNK activation; inhibits complex I from the respiratory chain at mitochondrial level; activates AMPK and Redd1, and inhibits mTOR, inducing oxidative stress [278] Decreases cell adhesion and invasion by inhibition of fibulin-3 and MMP2 expression [279]. Reduces STAT3 phosphorylation [280]. Decreases proliferation capacity by decreasing SOX2 expression [281]. Plus sorafenib: induces ROS, lipoperoxidation and inhibits efflux pumps, reducing proliferation and inducing apoptotic death [282]. Downregulates the levels of NF-κB and CREB via AMPK [283]. Plus As_2_O_3_: promotes autophagy and apoptosis through the activation of AMPK-FOXO3 via [284,285].	Plus surgery and TMZ: PFS in patients with GBM without metformin treatment was 6.77 months (range 1–54 months), OS was 14.58 months (range 1-84 months). Patients treated with metformin showed a significantly higher PFS (10.13 months, *p* = 0.018) [287]. OS in grade-III glioma: HR = 0.30, CI95% = 0.11–0.81; PFS: HR = 0.29, CI95% = 0.11–0.78 [288]. Plus TMZ and radiotherapy: did not significantly improve OS nor PFS [289]
**Tyrosine kinase inhibitors**		
**Erlotinib** 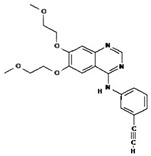 Synthetic compound; molecular formula: C_22_H_24_ClN_3_O_4_ [6–bis(2-methoxy ethoxy)quinazoline-4-yl]-(3-ethynylphenyl)amine]	Inhibits EGFR-dependent cell proliferation and induces arrest in the G1 phase of the cell cycle [293]. Plus CQ: induces cell death by activating caspase-independent pathways and the autophagic pathway [294]. Plus sorafenib: significantly increases cell death through accumulation of BIM, downregulation of Bcl-2 and Bcl-xL, and PARP cleaving; induces autophagy by accumulation of LC3-II and phosphorylation of mTOR [295]. Plus sorafenib, increased cell death by apoptosis and autophagy through of the inhibition of EGFR and RAF/MEK signaling [295]. Plus NSC23766 (a RAC1 inhibitor) inhibits cell proliferation by inducing autophagy and caspase 3-independent apoptosis with a decreased expression of Survivin [296].	Plus carmustine or TMZ: PFS-6 was 11.4% in the erlotinib group alone, and 24% in the carmustine or TMZ groups [299]. Plus TMZ, either before or after radiation therapy, increased PFS-6 from 14.1 months (historical control) to 19.1 months [300]. Plus bevacizumab mean PFS was 13.5 months vs. historical mean of 8.6 months; mean survival was 19.8 months vs. historical mean of 18 months [301]. Plus TMZ and radiotherapy (median age, 10 years; range: 3 to 19 years): treatment did not change disease prognosis [303].
**Gefitinib** 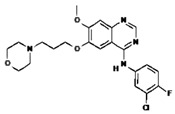 Synthetic compound; molecular formula: C_22_H_24_ClFN_4_O_3_ [*N*-(3-chloro-4-fluorophenyl)-7-methoxy-6-(3-morpholin-4-ylpropoxy)quinazolin-4-amine]	Binds ERBB-3, inhibiting the EGFR cell signaling, which induces arrest in the G1 phase of the cell cycle and inactivates the kinases ERK and AKT [305]. Induces apoptosis by dephosphorylating the Bad proapoptotic protein, with the ensuing BAX translocation into mitochondria and the activation of caspase 9 [307]. Inhibits cell proliferation, induces autophagy by activating the LKB1/AMPK signaling pathway and reduces tumor volume in xenotransplanted mice [310]. Plus MK-2206 (an AKT inhibitor): induces apoptosis (Survivin downregulation and Bim upregulation), and autophagy (LC3-II elevation) by inhibiting the phosphorylation of AKT/mTOR/S6K [307]. Plus valproic acid: induces autophagy by activating the LKB1/AMPK/ ULK1 signaling pathway, inhibiting cell proliferation and reducing tumor volume [310]. Plus β-elemene: induces apoptosis and autophagy by inhibiting the EGFR signaling [311].	Shows partial tumor regression [312]. No significant changes in patient survival were observed when administered as adjuvant for radiotherapy [313,314,315].
**Dacomitinib** 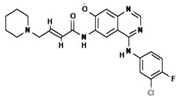 Synthetic compound; molecular formula: C_24_H_25_ClFN_5_O_2_ [(*E*)-*N*-[4-(3-chloro-4-fluoroanilino)-7-methoxyquinazolin-6-yl]-4-piperidin-1-ylbut-2-enamide]	Inhibits HER1, HER2 and HER4 [316]. Induces apoptosis in vivo [316]. Inhibits cell proliferation and reduced the activation of EGFR; promoting the inactivation of Akt and ERK1/2 with or without EGFRvIII [317]. Inhibits the EGFR signaling through inactivation AKT, ERK, and mTOR/S6, promoting apoptosis [316]. Cross the blood–brain barrier (BBB), reducing the intracranial tumor growth through the inhibition of EGFR signaling and self-renewal of carcinogenic stem cells [318]. Inhibits cell viability and induces apoptosis through inactivation of EGFR/PI3K/MAPK/mTOR pathway [319].	∙ Poor antineoplastic activity in patients with EGFR amplification, with or without mutation in EGFRvIII [320].
**Imatinib** 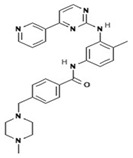 Synthetic compound; molecular formula: C_29_H_31_N_7_O [4-[(4-methylpiperazin-1-yl)methyl]-*N*-[4-methyl-3-[(4-pyridin-3- ylpyrimidin-2-yl)amino]phenyl]benzamide]	PDGFRα and PDGFRβ, and non-receptor kinase proteins (C-abl and BRCP) inhibitor [323]. Induces autophagy by increasing the phosphorylation of ERK1/2 and inhibiting the AKT/mTOR signaling [324]. Plus chloropyramine: enhanced anti-neoplastic activity through autophagy activation followed by apoptosis [325]. Plus carvedilol: induces autophagy, cell cycle arrest, and apoptosis due to generation of ROS and mitochondrial damage [327]. Plus nilotinib (a second-generation inhibitor of PDGFRα): increases cell migration and invasion through activation of p130Cas, FAK, and paxillin, without activity of PDGFRβ nor c-Abl [329].	Poor antitumor effect in glioblastoma patients, with a PFS-6 less than 16% and a radiological response below 6% [330]. Plus hydroxyurea: promising results, with a PFS-6 of 27% and a median PFS of 3.5 months, with a 42% of radiological response [332].
**Sunitinib** 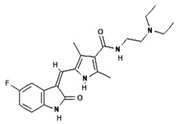 Synthetic compound: C_22_H_27_FN_4_O_2_ [*N*-[2-(diethylamino)ethyl]-5-[(*Z*)-(5-fluoro-2-oxo-1*H*-indol-3-ylidene)methyl]-2,4-dimethyl-1*H*-pyrrole-3-carboxamide]	Inhibits the activity of VEGFR2/3, PDGFRβ, FLT3, and c-KIT [333]. Reduces cell proliferation by promoting arrest in the G_2_/M phase of cell cycle; induces apoptosis through activation of caspase-3, and shows anti-invasive effect by reduction in activation of Src and FAK [334]. Decreases the accumulation of extravasated vascular accessory cells in the perivascular niche [335]. Activates autophagy by increasing the levels of Beclin-1, LC3-II, and autophagosomes formation [335]. Plus CQ: increased cytotoxicity and antiangiogenic capacity [338]. Plus CQ: induces apoptosis by increasing the levels of Bcl-2 and p53, and decreasing Bax [339]. Plus acetylsalicylic acid: shows anti-proliferative, anti-angiogenic, anti-invasive and pro-apoptotic effect through the inhibition in the release of VEGF, VEGFR1/2, HRAS, PI3K, AKT, MEK, ERK [338,340]. Plus gefitinib or sorafenib: inhibits neurosphere proliferation and growth by inhibition of AKT, MAPK, and STAT3 [341]. Plus miR-145 mimic: enhanced anti-neoplastic effect through downregulation of Cyclin D and increasing activity of P-gp and Bcrp [342]. Increases the antiproliferative and proapoptotic effect of TMZ in vivo but induces vascular resistance by increasing Ang-1 and Tie 2 [343]. EphB4 favors vascular resistance to sunitinib treatment by disturbing vascular morphogenesis, pericyte coverage, and inhibiting apoptosis in vitro and in vivo [345]. Induces over-expression of CXCR4 (G-protein-coupled receptor) and the small chemokine stromal-derived factor-1α (SDF1α) generating therapy resistance [347]. Plus PRX177561 (CXCR4 inhibitor): increased anti-angiogenic effect, decreasing tumoral growth and blood vessel density as well as augment the survival [348].	Patients with recurrent primary glioma failed to show an objective radiological response, with a PFS-6 of 12.5%, median PFS of 2.2 months, and median OS of 9.2 months [349]. Plus irinotecan and lomustine shown a low antitumoral response in recurrent glioblastoma patients [350]
**Sorafenib** 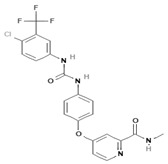 Synthetic compound: C_21_H_16_ClF_3_N_4_O_3_ [4-[4-[[4-chloro-3-(trifluoromethyl)phenyl]carbamoylamino] phenoxy]-*N*-methylpyridine-2-carboxamide]	Inhibits the activity of VEGFR2/3, PDGFRΒ, FLT3, and C-KIT. Inhibits the RAF/MEK/ERK pathway and downregulates the anti-apoptotic protein MCL-1 [351]. Inhibits cell proliferation and induces apoptosis through a downregulation in the activity of PI3K/Akt, MAPK, and MCL-1 [356]. Induces autophagy. [207]. Plus TMZ: the autophagy induced by sorafenib is reduced and promotes apoptosis through the downregulation of Raf, Hsp27 and Hsp72 expression, increasing endoplasmic reticulum stress [207]. Enhances the anti-tumoral effect f TTFields, enhancing apoptosis and autophagy by ROS generation and downregulation of markers related to epithelial to mesenchymal transition like vimentin and fibronectin [359]. Plus lapatinib (ERBB1/2/4 inhibitor): induces autophagy (ER stress) and apoptosis (via death receptors and mitochondrial) by the inactivation of ERK, AKT, MTOR, P70, and S6K, and decreasing the levels of MCL-1 and BCL-XL [360].	Plus temsirolimus: showed no benefit in survival of patients with recurrent glioblastoma [362]. Plus bevacizumab [365] and sorafenib with tipifarnib: showed advantages, but a high toxicity ratio [366].
**RAS/RAF/MAPK inhibitors**		
**Lonafarnib** 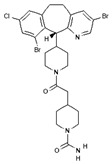 Synthetic compound: C_27_H_31_Br_2_ClN_4_O_2_ [4-[2-[4-[(2*R*)-6,15-dibromo-13-chloro-4-azatricyclo[9.4.0.0^3,8^]pentadeca-1(11),3(8),4,6,12,14-hexaen-2-yl]piperidin-1-yl]-2-oxoethyl]piperidine-1-carboxamide]	Farnesyltransferase inhibitor [368,369,370] Induces autophagy by inactivating the Ras/PI3K/AKT/mTOR pathway, with ROS formation and damage to DNA [368]. Inhibits cell proliferation and induces arrest in the G2 phase of the cell cycle by reducing pMAPK levels in glioma cell lines overexpressing EGFR [375].	Plus TMZ: showed a PFS-6 of 38% and a median PFS of 3.9 months. Median disease-specific survival was 13.7 months in 27% of recurrent glioblastoma patients [379].
**VEMURAFENIB** 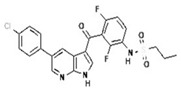 Synthetic compound: C_23_H_18_ClF_2_N_3_O_3_S [*N*-[3-[5-(4-chlorophenyl)-1*H*-pyrrolo[2-b]pyridine-3-carbonyl]-2,4-difluorophenyl]propane-1-sulfonamide]	RAF and ERK inhibitor [381]. Plus CQ: synergic effect to eliminate BRAFV600E-positive tumor cells, increasing cell death by caspase 3/7 activation [383].	Induces size reduction in metastatic glioma with a BRAF mutation (V600E) [383]. Plus everolimus: prevents disease progression over a period of 6 months until resistance emergence in patients with extracranial metastatic anaplastic oligoastrocytoma with a BRAF (V600E) and PIK3CA mutations [222].
**PI3K/AKT/mTOR Inhibitors**		
**Temsirolimus** 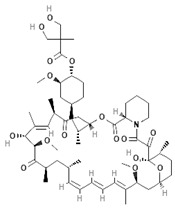 Synthetic compound: C_56_H_87_NO_16_ [(1*R*,2*R*,4*S*)-4-[(2*R*)-2-[(1*R*,9*S*,12*S*,15*R*,16*E*,18*R*,19*R*,21*R*,23*S*,24*E*,26*E*,28*E*,30*S*,32*S*,35*R*)-1,18-dihydroxy-19,30-dimethoxy-15,17,21,23,29,35-hexamethyl-2,3,10,14,20-pentaoxo-11,36-dioxa-4-azatricyclo[30.3.1.0^4,9^]hexatriaconta-16,24,26,28-tetraen-12-yl]propyl]-2-methoxycyclohexyl]-3-hydroxy-2-(hydroxymethyl)-2-methylpropanoate]	A rapamycin analogue that binds the FK-binding protein 12 (FKBP-12), inhibiting mTOR [394,395]. Shows an anti-proliferative and pro-apoptotic effect [396,398]. Modulates the activity of the NF-κB, PKC-α, and PI3K/AKT/mTOR pathways [399]. Plus PENAO (mitochondrial inhibitor): increases mitochondrial dysfunction, enhances cytotoxic activity inducing apoptosis and decreases clonogenic ability of diffuse intrinsic pontine glioma cells [402].	No antineoplastic effect better than TMZ in patients with unmethylated MGMT promoter in glioblastoma [397]. No antineoplastic effects, either as a monotherapy or combined with erlotinib in GBM patients [304]. Plus bevacizumab: showed radiological response (median PFS, 8 weeks; OS was 15 weeks) in recurrent GBM patients [409]. Plus sorafenib: did not improve survival of recurrent GBM patients [361].
**Everolimus** 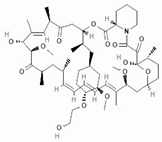 Synthetic compound: C53H83NO14[(1*R*,9*S*,12*S*,15*R*,16*E*,18*R*,19*R*,21*R*,23*S*,24*E*,26*E*,28*E*,30*S*,32*S*,35*R*)-1,18-dihydroxy-12-[(2*R*)-1-[(1*S*,3*R*,4*R*)-4-(2-hydroxyethoxy)-3-methoxycyclohexyl]propan-2-yl]-19,30-dimethoxy-15,17,21,23,29,35-hexamethyl-11,36-dioxa-4-azatricyclo[30.3.1.0^4,9^]hexatriaconta-16,24,26,28-tetraene-2,3,10,14,20-pentone]	Inhibitor of mTOR [396,417]. Decreases the expression and activity of lactate dehydrogenase and choline kinase α, reducing the levels of lactate and phosphocholine by transcriptional inhibition of HIFα in glioma models, both in vitro and in vivo [418]. Inhibits MCL-1 pro-apoptotic protein [420]. Plus palbociclib (CDK4/6 inhibitor): prevents DNA synthesis and induces apoptosis [422]. Plus LBT613 (a RAS inhibitor): inhibits cell proliferation, migration, and invasion [411]. Plus Delta-24-RGD adenovirus: induces autophagy by promoting the expression of Atg5; improves the mean survival of mice with glioma [425]. Enhances the effect of TMZ to induce cell death by autophagy [410]. Plus AZD2014 and Olaparib (an inhibitor of poly (ADP-ribose) polymerase): induces necroptosis [420].	Plus radiotherapy/TMZ and TMZ as adjuvant: well tolerated by GBM patients [426]. Despite increased toxicity, no improvement in patient survival was found [427,428]. Plus bevacizumab and radiotherapy/TMZ: median PFS of 11.3 months, median OS of 13.9 months in GBM patients [430]. Plus gefitinib: no therapeutic efficacy with respect to the historical control, even though a radiological response was observed in 33% of recurrent glioblastoma patients [431].

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
