# Peer review of "Autophagy as a Potential Therapy for Malignant Glioma"

_pharmaceuticals, 2020, doi:10.3390/ph13070156_

Round 1
Reviewer 1 Report
In this manuscript, the basic mechanisms of autophagy and the role of autophagy in glioma are both well written and discussed.
Author Response
Reply to reviewer 1:
In this manuscript, the basic mechanisms of autophagy and the role of autophagy in glioma are both well written and discussed.
Thank you for your comments.
Reviewer 2 Report
A manuscript of the review article entitled “Autophagy as a potential therapy for malignant glioma” (pharmaceuticals-843329) is an interesting, well-written and informative paper. Currently, the mechanisms of autophagy and potential gliomas therapy are still not fully elucidated. This review summarizes the current knowledge about the molecular crosstalk between autophagy and the carcinogenic mechanisms.
Comments and Suggestions for Authors:
- The abbreviation section should be the part of the main text (not a non-published section).
- Editorial comment - In the Table 1 (and the whole manuscript) the abbreviation miR or microrna should be standardize.
Author Response
Reply to reviewer 2:
Thank you for your comments.
A manuscript of the review article entitled “Autophagy as a potential therapy for malignant glioma” (pharmaceuticals-843329) is an interesting, well-written and informative paper. Currently, the mechanisms of autophagy and potential gliomas therapy are still not fully elucidated. This review summarizes the current knowledge about the molecular crosstalk between autophagy and the carcinogenic mechanisms.
Comments and Suggestions for Authors:
- The abbreviation section should be the part of the main text (not a non-published section).
The abbreviations were included in the main text, as suggested. (Lines 1628-1732)
- Editorial comment - In the Table 1 (and the whole manuscript) the abbreviation miR or microrna should be standardize.
This abbreviation was standardized in Table 1 and in the whole manuscript as miR.
Reviewer 3 Report
This is a very exhaustive review of autophagy in GBM. There are several recommendations:
- Almost every other sentence has a grammatical error! There needs to be a very thorough review by someone with English as their primary language. Do a spellcheck. Reduce unnecessary verbage. Focus on GBM data.
- Figure 1 is confusing. The top half of the figure should include all key stages as you describe: “induction, nucleation, elongation, and completion; fusion to lysosome, demotion and recycling.” It is unclear where the molecular pathways pictured in the lower half of the figure occur throughout these stages, so redesign the lower half of the figure to clarify.
- All tables need to be reconfigured so they can be turned clockwise 90 degrees and read like the rest of the manuscript.
- Recommend adding additional figures to illustrate the mechanisms of action (MOAs) of the specific agents discussed throughout the manuscript, specifically highlighting only those MOAs directly impacting autophagy.
- All of these pharmaceutical agents discussed have failed in clinical trials, greatly reducing enthusiasm for them. Recommend reducing/streamlining your discussion on all these failed agents. Instead, recommend adding sections devoted to NOVEL ways of targeting autophagy that have not been tested yet in clinical trials.
Author Response
Reply to reviewer 3:
Thank you for your comments and suggestion
This is a very exhaustive review of autophagy in GBM. There are several recommendations:
- Almost every other sentence has a grammatical error! There needs to be a very thorough review by someone with English as their primary language. Do a spellcheck. Reduce unnecessary verbage. Focus on GBM data.
The paper was revised by a native English-speaking editor.
- Figure 1 is confusing. The top half of the figure should include all key stages as you describe: “induction, nucleation, elongation, and completion; fusion to lysosome, demotion and recycling.” It is unclear where the molecular pathways pictured in the lower half of the figure occur throughout these stages, so redesign the lower half of the figure to clarify.
Figure 1 was redesigned. The left side of the edited figure shows the main signaling pathways that regulate autophagy, either positively or negatively, as well as the main points that induce autophagy. The right side of the figure shows the genic regulation of autophagy by transcriptional factors.
- All tables need to be reconfigured so they can be turned clockwise 90 degrees and read like the rest of the manuscript.
The tables were reconfigured and turned as suggested and now they read like the rest of the manuscript.
- Recommend adding additional figures to illustrate the mechanisms of action (MOAs) of the specific agents discussed throughout the manuscript, specifically highlighting only those MOAs directly impacting autophagy.
Figure 3 was added to the revised manuscript. It shows the therapeutic target of each drug, since their mechanism of action was widely described in the tables.
- All of these pharmaceutical agents discussed have failed in clinical trials, greatly reducing enthusiasm for them. Recommend reducing/streamlining your discussion on all these failed agents. Instead, recommend adding sections devoted to NOVEL ways of targeting autophagy that have not been tested yet in clinical trials.
A section on Novel and promising strategies under preclinical study was added to the revised manuscript (lines 1497-1588). Other compounds with antineoplastic activity currently under preclinical study are mentioned throughout the text, and we discuss promising alternatives to treat GBM, for which we have no cure yet, unfortunately. While these agents have not proved to be as efficient as expected in clinical studies, they are the most feasible approaches to be implanted in the clinical practice, either as a monotherapy or combined with other drugs.
Thanks to the reviewers for their valuable comments and suggestions, which certainly improved our manuscript.